# Modelling digital and manual contact tracing for COVID-19. Are low uptakes and missed contacts deal-breakers?

**Andrei C. Rusu** [1]*, **Rémi Emonet** [2☯], **Katayoun Farrahi** [1☯]

1 Vision, Learning and Control Research Group, University of Southampton, Southampton, United Kingdom,
2 Department of Machine Learning, Laboratoire Hubert Curien, Saint-Etienne, France

☯ These authors contributed equally to this work.
* ar5g15@soton.ac.uk

**Data Availability Statement:** All relevant data are to be found within the paper and the Supporting information files.

**Funding:** AR is an UKRI-funded PhD student. The funders had no role in study design, data collection

## Abstract

Comprehensive testing schemes, followed by adequate contact tracing and isolation, represent the best public health interventions we can employ to reduce the impact of an ongoing epidemic when no or limited vaccine supplies are available and the implications of a full lockdown are to be avoided. However, the process of tracing can prove feckless for highly-contagious viruses such as SARS-CoV-2. The interview-based approaches often miss contacts and involve significant delays, while digital solutions can suffer from insufficient adoption rates or inadequate usage patterns. Here we present a novel way of modelling different contact tracing strategies, using a generalized multi-site mean-field model, which can naturally assess the impact of manual and digital approaches alike. Our methodology can readily be applied to any compartmental formulation, thus enabling the study of more complex pathogen dynamics. We use this technique to simulate a newly-defined epidemiological model, SEIR-T, and show that, given the right conditions, tracing in a COVID-19 epidemic can be effective even when digital uptakes are sub-optimal or interviewers miss a fair proportion of the contacts.

## 1 Introduction

### 1.1 Problem overview

The epidemic started in Wuhan, China by the SARS-CoV-2 virus has uncontrollably spread through communities from all around the world, rapidly becoming a major global threat, which was responsible for more than 237 million infections and 4.8 million deaths by October 2021 [1]. Prompted by the scale of this epidemic, cross-disciplinary teams started working against the clock to develop reliable pathogen spreading models that could be used to assess the effectiveness of different public health interventions. Since imposing a general lockdown has proven economically unbearable for most countries, the attention significantly shifted to less restrictive yet partially successful measures, such as educating the public to socially distance, deploying large-scale testing and quarantining contacts through various tracing

and analysis, decision to publish, or preparation of the manuscript.

**Competing interests:** The authors have declared that no competing interests exist.

mechanisms [2]. The latter proved rather challenging for the traditional interview-based approaches, mostly due to significant delays, staffing issues and a generally poor recollection exhibited by the interviewees. As a result, digital alternatives were quickly sought after by several governments. These were or are currently being deployed in many states, most of them being reliant on either a Bluetooth solution, such as the Exposure Notification (GAEN) system [3], or a geolocation-based software, similar to the Integrated Disease Surveillance Programme in India [4]. That being said, the efficiency of these strategies remains largely dependent on the application adoption rates and the behavioral patterns of their userbase (i.e. self-isolation compliance, respecting the usage guidance, keeping the tracing device turned on etc.). Although some have suggested an application uptake of at least 50% would be needed at the population level to contain the epidemic [5], others showed via simulations that 60% would be enough to stop the spread without requiring further interventions [6]. That being said, the adoption levels generally quoted in the literature as "sufficient" remain mostly unattainable due to privacy concerns and internet access limitations. The picture gets even more intricate when the aforementioned behavioral issues are widespread in the users' communities or if inadequate testing regimes and manual tracing procedures are employed.

Motivated by the limited evidence we have of the efficacy exhibited by contact tracing methods in the face of such challenges, we developed a *multi-site mean-field* model that can simulate the joint effects of these variables on the evolution of an epidemic, and used it to study COVID-19 via a new disease-specific compartmental formulation—SEIR-T. Our methodology draws inspiration from the work of [7], but it enables the simulation of more varied scenarios involving digital tracing at different uptake levels $r$, manual tracing with various network overlaps $\Gamma$, or both procedures combined. Additionally, we propose separating the "traced" status from the infection states, thus allowing for a node to get isolated at all times (unless it has reached an end state, i.e. recovered or dead), while also ensuring self-isolation can end due to non-compliance or term expiration, all without impacting an individual's standard disease progression. This feature also makes our approach directly compatible with any other compartmental model. As is customary, all our code was made publicly available (see S1 File).

The experiments we conducted confirm that the potency of contact tracing not only depends on the accuracy of the tracing network, but also on several other variables (i.e. testing rates, tracing reliability, staffing and delays, public-health communiqués, isolation conformity etc.), an optimal configuration of these given a country's epidemiological situation being essential for a swift viral containment. Even when lower uptakes are registered ($r < 0.4$) or the interviewing process misses many contacts ($\Gamma \leq 0.5$), our simulations suggest that significant reductions in the peak of infections and the total number of deaths can still be achieved given small tracing delays and the appropriate levels of testing and self-isolation compliance. What is more, the combined effects of manual and digital tracing can drive the effective reproduction number $R$ below 1 even when neither is very efficient. We validate our results on numerous parameter configurations and several classic network topologies, including random Erdős–Rényi [8], scale-free [9–11], small-world [11–13], and a real social network [14].

## 1.2 Related work

In recent years, modelling epidemics has mainly been achieved via either of two paradigms: agent-based (ABM) or equation-based models (EBM). The first represents a bottom-up approach in which a set of behaviors are attributed to each agent in a topological system. These behaviors dictate every individual's action patterns through this topology, and ultimately

determine the execution of different discrete interaction events (e.g. infection spreading, tracing notification broadcasting etc). ABMs tend to be relatively complex and resource-intensive to simulate, the involved cost being often justified by their exhibited level of granularity and their ability to monitor public interventions at the individual level [15]. Government-advising groups in the UK decided to employ this paradigm early on in the COVID-19 pandemic to estimate the effects of such interventions [16, 17]. A more recent Oxford study looked at the combined effects of manual tracing with digital solutions, at various application uptakes, via a rich yet scalable ABM fitted to mobility data from different counties in Washington [18]. We consider their findings the strongest modelling evidence to date that digital tracing can be effective even at low adoption rates.

On the other hand, EBMs define a set of equations that express the evolution of certain *continuous* observables over time. These generally represent system states (called compartments) showing how a disease progresses through a population. The SIR process, a widely-known EBM, utilizes three ordinary differential equations to model a generic epidemic [19]. Extensions of SIR were subsequently used to simulate the transmission of many pathogens, including Zika [20], Ebola [21], and most recently SARS-CoV-2—e.g. SIDARTHE [22], SUQC [23]. The present study employs a variation of the compartmental model designed by the French National Institute of Health and Medical Research (Inserm) to study the impact of lockdown exit strategies on the spread of COVID-19 [24].

At the intersection of these two paradigms lies the category of multi-site mean-field models which combine the mathematical rigour and the superior generalizability of EBMs with the ability to leverage locality information regarding every individual. Similarly to ABMs, the infection spreads over a predefined network that can either be random [25] or inferred from real data [7], yet unlike ABMs, the dynamics are fully characterized by state transition equations. Huerta and Tsimring first employed this technique for modelling contact tracing in a generic epidemic [26, 27]. Farrahi et al. took the idea a step further by restricting the tracing propagation to a subset of the infection network, thus accounting for the inherently noisy nature of this process [7]. Even though both of these exhibited powerful modelling capabilities, they were limited by their underlying compartmental formulation (SIRT) which made several unrealistic assumptions that do not generalize to real viral diseases: inter alia, the recovery was conditioned on tracing, susceptibles could not be wrongfully isolated, a traced person remained noninfectious for the full duration of the epidemic. Our modelling approach fixes these issues by separating the traced/isolated status from the infection state, therefore allowing for all the "active" nodes (i.e. not hospitalized, recovered or dead) to become traced or exit self-isolation after a certain amount of time without changing their corresponding disease progression. Concurrently, this modification enables one to simulate the effects of contact tracing, independently of the compartmental model used.

For the sake of completeness, we would also like to mention that branching process models for epidemics have become increasingly popular in the last few years [28, 29]. One such model, concerned with studying the effects of manual contact identification together with digital tracing solutions at various uptakes on the COVID-19 pandemic, has recently been proposed [30]. Simulations conducted with it show that effective manual tracing needs to be coupled with an application uptake of at least 75% to achieve containment, although smaller adoption rates can decrease the reproduction number *R* if combined with other public health interventions. Our results are in accordance with the latter observation, but they also show that, given the right testing and tracing regimes (including good self-isolation compliance), lower and achievable adoption levels are actually enough to significantly reduce the viral spread, subject to the social network's connectivity patterns.

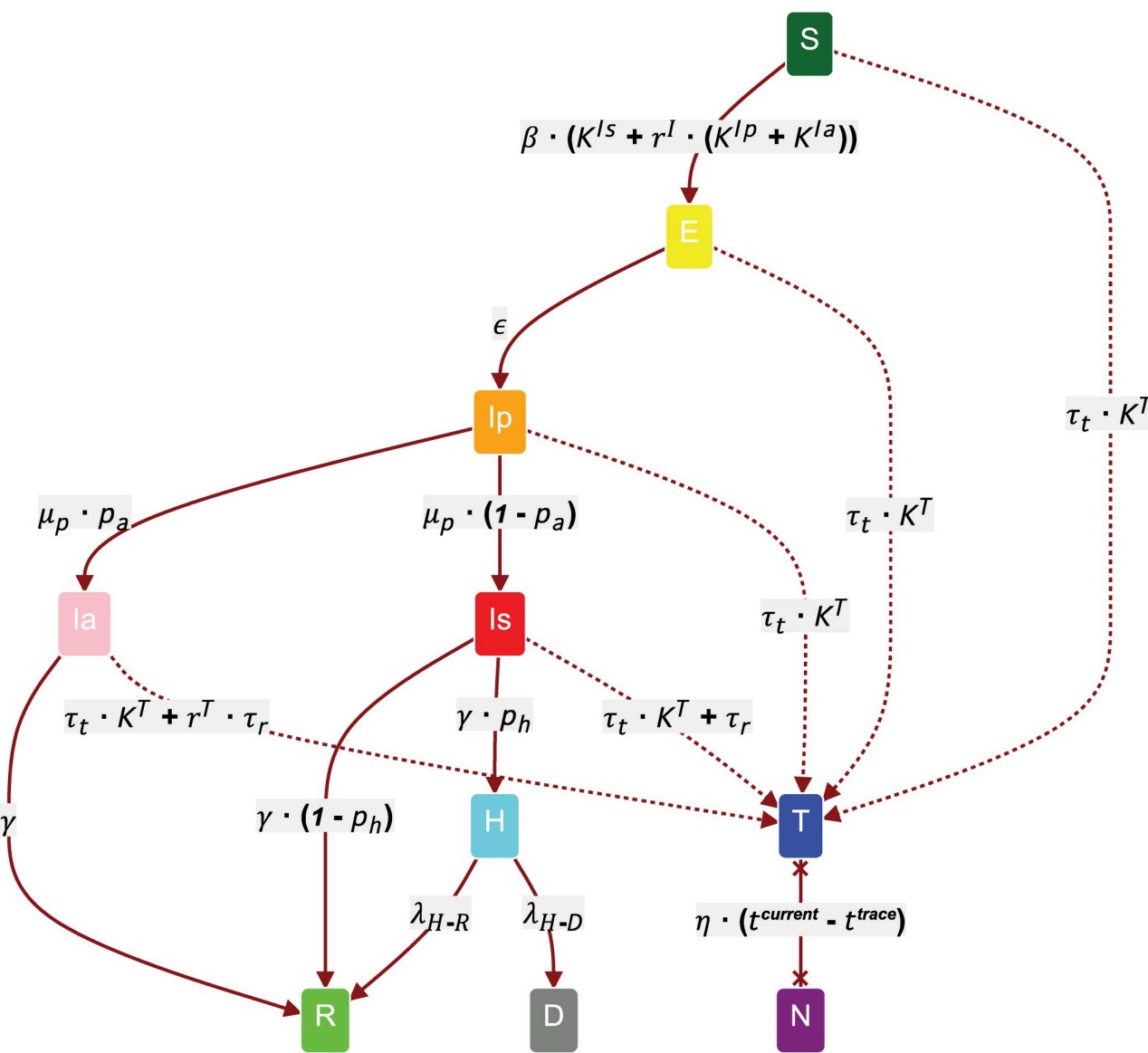

**Fig 1. The SEIR-T compartmental model for COVID-19.** Each node has 2 allocated variables: an infection state and a tracing status. The infection states from top to bottom are: $S$—susceptible; $E$—exposed but not infectious; $I_p$—infectious, presymptomatic; $I_a$—infectious, asymptomatic; $I_s$—infectious, symptomatic; $H$—hospitalized; $R$—recovered / removed; $D$—dead. At any point in time, a node's tracing status can either be $T$ (traced and isolated) or $N$ (not traced/isolated or non-compliant). Each state transition has a certain time-dependent probability $p_{S1 \to S2}$; the edge labels here represent both $\frac{p_{S1 \to S2}}{\Delta t}$, and the $\lambda$ rate of the corresponding exponential to sample from in the continuous-time simulations.

## 2 Materials and methods

### 2.1 Compartmental model outline

Motivated by growing evidence that simple SIR frameworks are inefficient at capturing the dynamics of SARS-CoV-2 epidemics [31], we developed a new compartmental model that accounts for many of its particular features. Each state transition represented in Fig 1 model is labeled with its corresponding time-dependent probability, an end configuration being reached when all non-susceptible nodes become either recovered (R) or dead (D). A

**Table 1. Compartmental model parameters.**

| Parameter | Value(s) | Description |
|---|---|---|
| $\beta$ | 0.0791 | Transmission rate corresponding to $R_0 = 3.18$. According to maximum likelihood estimation performed by [24]. |
| $K^X$ | $\mathbb{R}$ | Function mapping nodes to the total number/weight of connections to neighboring nodes in state $X \in \{I_p, I_a, I_s, T\}$ for a given network. |
| $r^I$ | 0.5 | Relative infectiousness of $I_p$ and $I_a$ compared to $I_s$. This is still disputed: $\approx 0.5$ according to [24, 32], but weak evidence as per [33]. |
| $\epsilon^{-1}$ | 3.7 | Latency period, measured in days. Source: [24]. |
| $p_a$ | 0.2 / 0.5 | Probability of being asymptomatic. This is still disputed: 0.2 used by [24, 34], but 0.5 according to [35, 36]. |
| $\mu_p^{-1}$ | 1.5 | Presymptomatic period, measured in days. Source: [37]. |
| $p_h$ | 0.1 | Probability of being hospitalized for *adults* (can be considerably different for children/seniors). Equivalent to $p_{ss}$ in [24]. |
| $\gamma^{-1}$ | 2.3 | Infectious period considering the mean generation time 6.6 days. Source: [24]. |
| $\lambda_{H-R}$ | 0.083 | Daily rate of recovery for *adults* (different for children/seniors). Source: [38]. |
| $\lambda_{H-D}$ | 0.0031 | Daily rate of deaths for *adults* (different for children/seniors). Source: [38]. |
| $\tau_t$ | [0–0.5] | Contact tracing rate. Encompasses multiple related phenomena: the tracing latency/ efficiency due to staffing/server reliability, depending on the type of tracing; the likelihood of remaining isolated given the number of traced neighbors. Ranges from no tracing (0) to every 2 days on average (0.5). |
| $\tau_r$ | (0–0.5] | Testing / Random tracing rate. Ranges from almost no testing (0.001) to every 2 days on average (0.5). |
| $r^T$ | 0.8 | Relative probability for $I_a$ to be tested positive (against $I_s$). Assume testing $E$ and $I_p$ rarely happens or results in false negatives most of the time. |
| $\eta$ | 0 / 0.001 | Non-compliance / Self-isolation exit rate. Scaled by the time elapsed since beginning the isolation: $t^{current} - t^{trace}$. |

description of the model parameters, together with the values we consider for each of them, can be consulted in Table 1.

## 2.2 Network propagation mechanism

Our propagation model consists of a predefined network on which the infection spreads, and one subnetwork ascribed to each type of contact tracing (manual or digital). This mechanism allows for simulating either one tracing strategy in isolation (*dual* topology, example in Fig 2) or both in tandem (*triad* topology, Fig 3). Connected vertices in the true infection network are to be considered "close contacts", as defined by institutions like the CDC [39].

The tracing graphs are usually subset views of the true contacts network, where missing edges correspond to application misuse in the digital setting or contacts not recalled in the manual interviewing process, while isolated vertices are used to represent individuals that never run a government's digital solution or are effectively unreachable. Be that as it may, people can at times overestimate the number or the duration of their social interactions [40], and thus it is possible that tracers are occasionally pursuing erroneous links. Even though our model can simulate "false" contacts, similarly to [7], we consider their occurrence quite rare during a global pandemic (and thus negligible), since the public health personnel is particularly well-trained and the general public is more attentive. We control the subsetting of the infection graph via two interlinked parameters: the *degree of overlap* $\Gamma = \frac{K - Z_{rem}}{K}$ and the *uptake rate* $r = \frac{N - N_{utn}}{N}$, variables which ultimately determine the values of $N_{utn}$ and $N_{ute}$ (refer to Eq 1). To be more explicit, the inputted $\Gamma$ and the infection network's mean degree $K$ are utilized to calculate $Z_{rem}$, the average number of edges per node to get removed from a tracing view. The

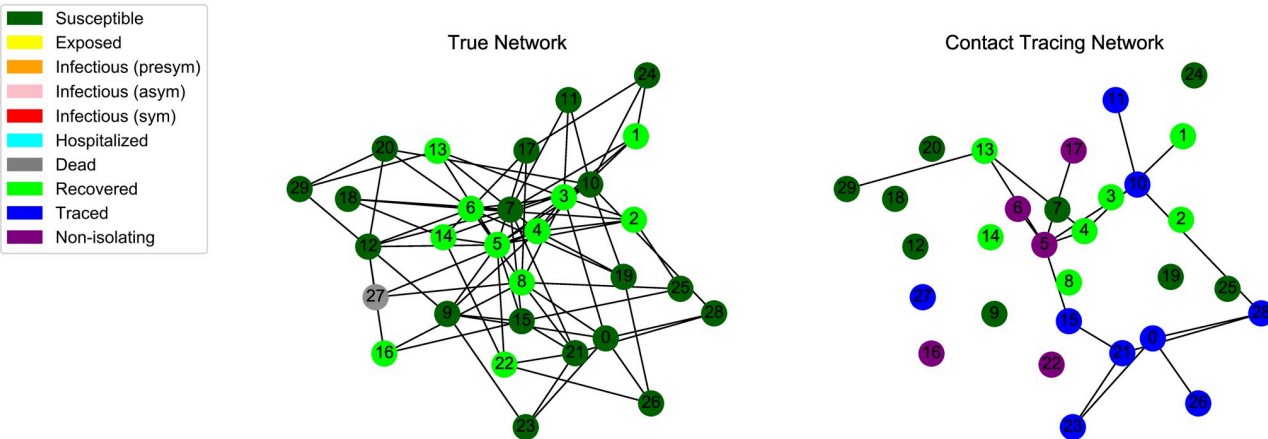

**Fig 2. Final state of an epidemic simulation over a *dual* topology.** Infection spreads with respect to the neighborhoods of the first network (here a SF graph); the second network corresponds to digital tracing at uptake $r = 0.5$.

latter effectively corresponds to marking as untraceable $N_{ute} = \frac{N \cdot Z_{rem}}{2}$ of the edges in the interaction graph. Similarly, the selected $r$ and the total number of nodes $N$ are used to establish how many vertices are to be made completely untraceable in a particular tracing subnetwork: $N_{utn} = N \cdot (1 - r)$. This work showcases simulations in which the first of these two parameters describes the accuracy of manual tracing, whereas the second quantifies the adoption of a digital solution. That being said, our model supports exploring more complex scenarios, where both the overlap and the uptake can be varied for a single tracing view. A full description of the network-related variables involved in our modelling procedure can be consulted in Table 2.

$$N_{utn} = N \cdot (1 - r) \quad N_{ute} = \frac{N \cdot Z_{rem}}{2} = \frac{N \cdot K \cdot (1 - \Gamma)}{2} \tag{1}$$

Throughout our experiments, we assume a "traced" individual (i.e. in state $T$) automatically enters self-isolation, so infecting or getting infected remains impossible until it becomes "non-

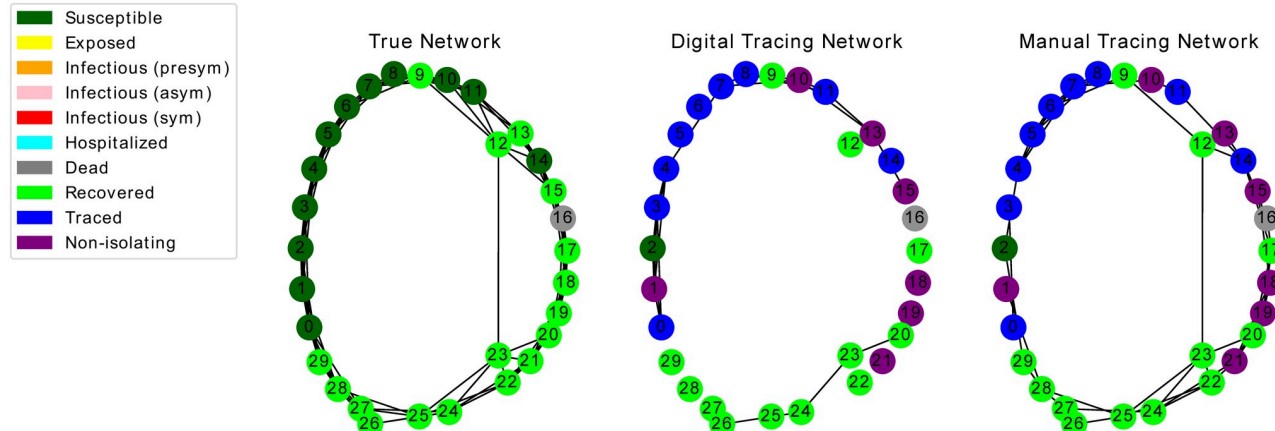

**Fig 3. Final state of an epidemic simulation over a *triad* topology.** Infection spreads with respect to the neighborhoods of the first network (here a SW graph); the second network corresponds to digital tracing at uptake $r = 0.5$, while the third involves manual tracing with overlap $\Gamma = 0.5$.

**Table 2. Network-generation parameters.**

| Parameter | Value(s) | Description |
|:---:|:---:|:---|
| $N$ | $\mathbb{N}$ | Population size of the infection network. |
| $K$ | 10/20 | Average degree of the infection network. We fix this for ER graphs. |
| $m$ | 10 | Random edges to add for each new node in Holme-Kim networks [11]. |
| $p_\triangle$ | 0.2 | Probability of completing a triangle after adding a random edge. |
| $\Gamma$ | [.1, 1] | Degree of overlap between infection network and a tracing subgraph. Used to calculate $Z_{rem}$, which in turn gives $N_{ute}$ (number of untraceable links). |
| $r$ | [.1, 1] | Uptake rate (between infection network and a tracing subgraph). Used to calculate $N_{utn}$ (number of untraceable nodes). |

isolating" ($N$). This can happen either legally (after 14 days) or unlawfully (with a probability of $\eta$ scaled by the time elapsed since isolating). In addition, we presume that a node's probability to get infected proportionately increases with the amount of infectious neighbors it has in the contacts network, while the likelihood of being traced and compliant with self-isolation recommendations is directly proportional to the number of adjacent $T$ nodes it features in each tracing subnetwork.

## 2.3 Simulation overview

The baseline simulations in this study were run over Erdős–Rényi random graphs (ER), featuring different population sizes and average degrees. It is worth mentioning that, although the epidemiological literature has widely adopted it, this type of graph model tends to be unsuitable for capturing the interaction patterns of many real social networks [41]. In spite of this, Tsimring and Huerta concluded that the SIRT-induced epidemic dynamics stays "qualitatively similar" between ER and the empirically motivated class of small-world graphs, since the realizations of both these models feature a well-defined epidemic threshold [27]. This result should also hold for our framework, considering that we model tracing in a fairly analogous fashion. What is more, the ER's inherent ability to accommodate the characteristics of randomly mixed populations [42] makes it an adequate vehicle for studying outbreaks in public places, such as stores or mass transit conveyances [18]. Random mixing models, in turn, were shown to offer acceptable estimates of the total epidemic size when the transmission probability is high or the infectious period is relatively small [43], conditions that are usually satisfied in the case of COVID-19 breakouts. Nonetheless, several experiments involving more realistic small-world (SW) and scale-free networks (SF) are the focus of a more detailed exploration in Secton 3.4. We note here that ABM simulations, mobility or contact tracing datasets could be utilized in conjunction with the configuration model to obtain even more accurate predictions for particular lifelike scenarios [44, 45], but these do not offer any generalization guarantee.

It is a known fact that the SARS-CoV-2 virus is an overdispersed infectious agent [46, 47], and like many other pathogens with a high epidemic potential [48], the disease diffusion is largely driven by "superspreading" events [49]. As such, SFs like the Barabási-Albert networks [9] tend to offer a sounder representation of the transmission chain since superspreaders can be adequately modelled as hubs in a specific social graph [50, 51], while the latter naturally arise as a result of the preferential attachment process that underpins SFs. On the other hand, SWs more closely resemble interactions in social networks due to their larger clustering coefficient, while clusters, in turn, have been shown to be an important catalyst of the COVID-19 pandemic [52]. We believe our modelling technique of preferentially sampling nodes with a higher traced neighbor count for undergoing quarantine to be similar in nature to the frequency-based contact tracing procedure employed in [51], and thus we expect superspreaders

inside SFs to get swiftly targeted by our control framework, subject to the strength of the contact tracing rate. Moreover, as we mentioned earlier, the tracing-imbued epidemic dynamics over SWs is akin to that of ERs, and thus similar levels of tracing efficiency are to be expected for both these graph models. It is therefore sensible to assume that the modelling mechanism we employ remains suitable for assessing the effects of tracing over a broader range of network types (other than ERs).

In contrast to the above, Secton 3.5 investigates the effects of digital and manual tracing in a viral outbreak simulated over a real social network, representing a tightly-connected community of 74 students and graduates from MIT who agreed to have their location and interactions monitored via WLAN and Bluetooth scans over an entire academic year (detailed exploratory analyses of the dataset can be examined in [7, 14]). In our simulations, this dynamic network changes daily over a period of 31 weeks, its links being *weighted* by the aggregated number of Bluetooth proximities recorded between their corresponding corner points on each particular day. In the static settings presented thus far, $K^X$ represents a function mapping nodes to the total number of neighbors in state $X \in \{I_p, I_a, I_s, T\}$ (see Fig 1). To account for dynamic weights, however, all $K^X$ terms get replaced by a time-dependent function $K_t^X$, given by Eq 2, where $K_{norm} = 10$ is a normalization factor that ensures the average function value remains above 1, $w_t^X(n)$ is the sum of edge weights incoming from those neighbors of node $n$ which are in state $X$ at time $t$, while $< W >$ is the overall static average weight. The latter represents an average over days of the average total weight per node, calculated using Eq 3, where $D = 216$ is the number of days within the considered 31-week period, $N = 74$ is the number of nodes for which we have contacts data, and $w_t(n)$ is the *total* weighted degree of node $n$ at time $t$ (i.e. irrespective of state).

$$K_t^X(n) = \frac{K_{norm} \cdot w_t^X(n)}{< W >} \tag{2}$$

$$< W > = \frac{\sum_t^D \sum_n^N w_t(n)}{D \cdot N} \tag{3}$$

In this work, the time intervals between two state changes of the same kind are assumed to form an exponential distribution, with the $\lambda$ rate equal to the corresponding transition label displayed in Fig 1. Choosing this distribution for timing the infection propagation, in particular, keeps our approach in line with many previous epidemiological works relying on compartmental formulations [7, 53, 54], while also being in accordance with the findings of different cohort studies involving wearable tracking devices that reported roughly-exponential decays in their participants' histogram of interactions [55, 56]. Similar cohort studies found heavier-tailed distributions based on power laws to be more compatible with the time intervals between successive interactions, citing the bursty nature of social dynamics as the determining factor [57, 58], yet the corresponding data fit was often imperfect while extensive comparisons against exponentials were not performed. In the epidemiological setting, several authors have argued for a shift towards more realistic and flexible Gamma (more commonly Erlang [59, 60]) or Weibull distributions [61–63] for the infection waiting times, emphasizing the non-Markovian behavior that epidemics occasionally exhibit. That being said, exponentials have been shown to provide a particularly good fit to epidemiological data when the mean generation time is correctly fixed [59] or the mean infection duration is smaller [64]. Both of these conditions constitute sensible assumptions in our case.

For efficiency, we simulate the COVID-19 outbreaks using Gillespie's algorithm [65], which has been shown to be stochastically exact to and faster than the Monte Carlo method

(MC) for both static and dynamic network-based diffusion processes [66]. Compared to a continuous-time MC simulation, which entails sampling the next transition for all the possible state changes, discarding all but the most "recent" event [56], Gillespie's procedure directly draws the time elapsed until the next transition and identifies the state change most likely to have taken place within that period. A detailed pseudocode for event sampling in our work is provided in Fig 4.

## 2.4 Metrics under consideration

Aside from scrutinizing the number of individuals in each compartment over time (please also refer to S1 File for more such evaluations), we assess the efficacy of different contact tracing strategies ($C_{\theta;\tau_t}$, under different $\tau_t$) by looking at their achieved *peak suppression* ($P_{sup}$) throughout all our simulations, thus comparing them against the corresponding no-tracing scenario ($C_{\theta;\tau t} = 0$) in which all parameters $\theta$ (but $\tau_t$) are left unchanged. Mathematically, this can be expressed through Eq 4, where $I_{max}$ is a function mapping parameter configurations $C_\theta$ to the average peak of infections recorded across multiple runs.

$$P_{sup} = I_{max}(C_{\theta;\tau_t=0}) - I_{max}(C_{\theta;\tau_t}) \tag{4}$$

Since the inception of the COVID-19 pandemic, the majority of the literature on epidemiological modelling and public-health messages alike have scrutinized different nonpharmaceutical interventions in relation with their impact on $R$, the effective reproduction number [24, 67]. For the latter more realistic scenarios (i.e. Sections 3.4 and 3.5), we also estimate the $R$ value after $t = 7$ days since $t_0$. To do so, we input the recorded exponential growth rate $\lambda$ to Eq 5, thus following the Wallinga and Lipsitch methodology [68]. The generation time distribution for our SARS-CoV-2 epidemics is assumed to be $Gamma(\alpha = 1.87, \beta = 0.28)$ [69], its moment-generating function being denoted with $M(.)$. To calculate $\lambda$ from the incidence rate $c(t)$ recorded within the time window $[t_0, t_0 + t]$, we use Eq 6 together with the initial number of infected $c(t_0)$.

$$R = \frac{1}{M(-\lambda)} = \frac{1}{(1 - \frac{-\lambda}{\beta})^{-\alpha}} = (1 + \frac{\lambda}{\beta})^\alpha \tag{5}$$

$$c(t) = c(t_0)e^{\lambda t} \Rightarrow \lambda = \frac{\log c(t) - \log c(t_0)}{t} \tag{6}$$

## 3 Results and discussion

### 3.1 Variation induced by population size

Initial simulations using ER graphs suggested the degree of variability across runs scales with the number of nodes. In order to verify this hypothesis, we design an experiment in which we vary the population size: $N \in \{200, 500, 1000, 2000, 5000, 10000, 20000\}$, while keeping the other parameters fixed at: average degree $K = 10$, *dual network* tracing with uptake $r = 0.5$ (overlap $\Gamma$ implicitly derived), asymptomatic probability $p_a = 0.2$, contact tracing rate $\tau_r = 0.1$, and testing rate $\tau_r = 0.1$, with one infectious individual set for time $t_0$. We note that, as $\tau_r = \tau_t > \beta$, contact tracing is expected to engulf the infection percolation in the limit. However, by choosing an uptake value considerably smaller than 1, we ensure our variance analysis remains significant since many of the randomly-generated tracing views end up producing a much slower discontinuation of otherwise quickly-contained infection cascades. This results in a

---

**Algorithm 1** Infection event sampling via Gillespie's algorithm adaptation

---

1: **global variables**
2:   $t$                                      ▷ Current global time of the simulation
3:   $n_u$                                    ▷ Node ID of the last transition update
4:   $N_I$                                         ▷ The infection contacts network
5:   $N_T$                                  ▷ List of tracing networks (IDs 0 and 1)
6:   $I_f$                          ▷ Getter of possible infection propensity functions
7:   $T_f$                            ▷ Getter of possible tracing propensity functions
8:   $\lambda$                        ▷ Dict mapping nodes to possible transitions (rate, $S_{to}$)
9: **end global variables**
10: **procedure** SAMPLE_NEXT_EVENT
11:     ▷ Collect nodes that need rates updating: last and neighbors
12:     $update\_nodes \leftarrow N_I.\text{neighbors}(n_u) \cup \{n_u\}$
13:     ▷ Update $\lambda$ for update_nodes
14:     **for each** $n \in update\_nodes$ **do**
15:         $\lambda.\text{pop}(n)$                          ▷ Invalidate node in the dict of rates
16:         $S_I \leftarrow N_I.\text{state}(n)$                      ▷ Get current infection state
17:         $S_T \leftarrow N_T.\text{state}(n)$                       ▷ Get current tracing state
18:         ▷ Update rates based on node state and the true network
19:         **for each** $(f_r, S_{to}) \in I_f(S_I)$ **do**              ▷ $f_r$ is a rate function
20:             ▷ rate based on neighborhood of $n$ in $N_I$ and a scalar
21:             $r \leftarrow f_r(N_I, n)$
22:             ▷ Add (rate, state) to the $\lambda$ dictionary of rates
23:             $\lambda(n) \leftarrow \lambda(n) \cup \{(r, S_{to})\}$
24:         **end for**
25:         ▷ Update rates based on tracing state and 2 tracing nets
26:         **for each** $(f_r, S_{to}) \in T_f(S_T)$ **do**
27:             ▷ rate $r_i$ depends on tracing neighborhood of $N_T(i)$
28:             $r_0, r_1 \leftarrow f_r(N_T(0), n), f_r(N_T(1), n)$
29:             $\lambda(n) \leftarrow \lambda(n) \cup \{(r_0, S_{to}), (r_1, S_{to})\}$
30:         **end for**
31:     **end for**
32:     ▷ Convert $\lambda$ into 2 lists related by the map rate $\leftrightarrow (node, S_{to})$
33:     $rs, nts \leftarrow \text{convert}(\lambda)$
34:     ▷ Gillespie sampling of the minimum exponential time
35:     $\lambda_{min} \leftarrow \sum_{r \in rs}(r)$
36:     $t_{min} \sim Exp(\lambda_{min})$
37:     ▷ Next time point is the current $t$ + the minimum sampled
38:     $t_u \leftarrow t + t_{min}$
39:     ▷ Categorical sampling of the actual transition
40:     $id_u \sim P(X = k) = \frac{rs(k)}{\lambda_{min}}$               ▷ Base rate over sum of rates
41:     $n_u, S_u = nts(id_u)$                          ▷ Change last-updated node
42:     $S_{past} = N_I.\text{state}(n_u)$
43:     ▷ Create Event $e$ dictionary which shall be used to update the network
    states, neighborhood counts and the epidemic stats
44:     $e = \{("id", n_u), ("from", S_{past}), ("to", S_u), ("time", t_u)\}$
45:     **return** $e$
46: **end procedure**

---

**Fig 4. Pseudocode for event sampling in the SEIR-T model.** We adapt Gillespie's algorithm for our network-based simulations, thus sampling the minimum event time directly. The list of rates is updated at each iteration only for the last updated node and its neighbors. The procedure returns an event dictionary which is then used to update the network states, neighbor counts and running statistics.

high probability for finite outbreaks to occur during the early stages of the simulations (i.e. above the epidemic threshold for enough time).

The statistics in Fig 5 size represent averages over several simulations conducted with each of the 10 different network initializations picked by a random sampler, filtering out those

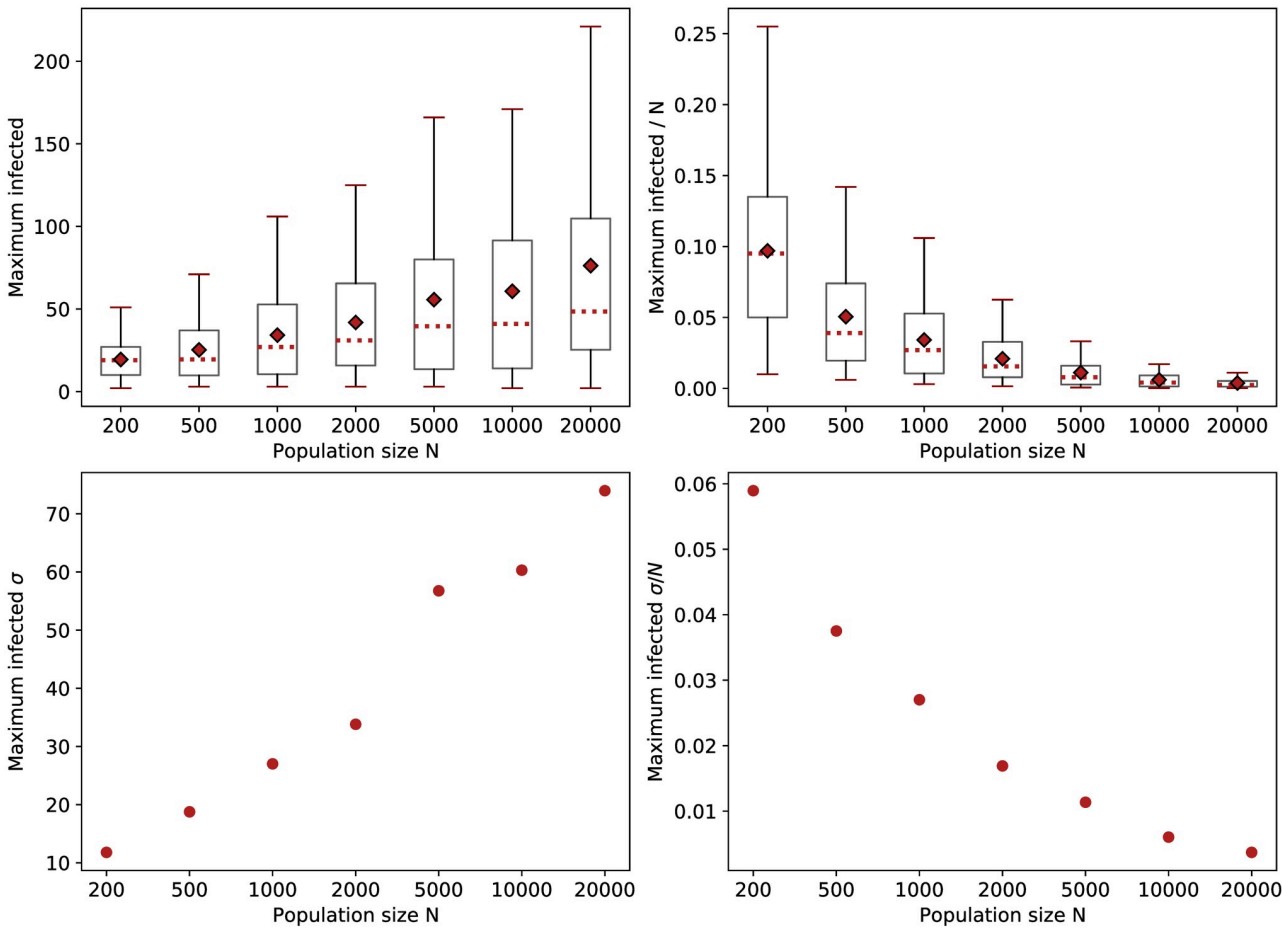

**Fig 5. Uncertainty of simulation results with regard to the infection peak.** Values from 80–100 runs plotted for different population sizes, $K = 10$, $\tau_t = \tau_r = 0.1$, $p_a = 0.2$. On top, boxplots with quartiles represented via whiskers, medians via red dotted lines, and averages via red diamonds; the standard deviations $\sigma$ are given below. The left-hand side displays absolute values, whereas on the right all the variation levels are scaled down by $N$.

iterations which registered less than three overall infected (for a total of 80–100 simulations overall per each value of $N$). The data confirm the variance in peaks of infection increases as the network expands, aspect which can be explained by the growing difference between early-stopped and full-blown outbreaks. In contrast, the uncertainty in estimating the relative percentage of these maximal points expectedly decays with the size ($\approx$ according to $\frac{1}{\sqrt{N}}$), a choice of $N = 1000$ resulting in a tolerable standard deviation of almost 3%, while $N = 10000$ leads to an even smaller variability of <1% across runs. Consequently, we consider these two values representative for our model's expressive power given a randomly-mixed population, and, as such, we use them both in our experiments. We account for the corresponding difference in variances by simulating 7 different networks with 15 random seeds each for $N = 10000$, but 50 networks and 15 seeds in the case of $N = 1000$. We note this result may not hold in the case of structured populations, yet nevertheless we reproduce the latter of the two setups in the SF-SW experiment for consistency.

### 3.2 Tracing overlap in larger populations

Going forward, we want to assess the effect of varying a tracing network's accuracy (i.e. overlap) in an outbreak involving a large community of $N = 10000$ individuals. To achieve this, we

use the following parameter configuration: an average degree $K = 10$, *dual network* tracing with overlap $\Gamma \in \{0.11, 0.22, \ldots, 1\}$ (uptake $r$ is implicitly derived), asymptomatic probability $p_a = 0.2$, a tracing rate $\tau_t \in \{0.01, 0.04, 0.07, 0.1\}$, a testing rate $\tau_r \in \{.001, 0.01, 0.04, 0.07, 0.1\}$, and a non-compliance rate $\eta = 0$ (assuming everybody self-isolates until they are no longer infectious), with a single $I_p$ node sampled at time $t_0$. The resulting statistics get averaged over 105 runs, as previously described.

Fig 6 shows that a sub-optimal test rate, such as $\tau_r = 0.001$, leads to inconclusive results, where the variance induced by the stochasticity of the process shadows any benefit obtained through contact tracing. With better testing, clearer patterns start to emerge: The higher the contact tracing rate, the better the peak suppression is and the faster it gets approached (see Fig 7). As $\tau_r$ becomes even more effective, smaller tracing network overlaps are needed to swiftly reduce that maximum point. Looking at the tracing rate, a moderate value of $\tau_t \in \{0.04, 0.07\}$ achieves a delay in the peak for smaller $\Gamma$, but this can occasionally lead to a prolonged epidemic, especially for overlaps in the "noise" region like $\Gamma = 0.11$, since initially-uninfected regions may get incorrectly traced, so the epidemic has the chance to gain momentum once those individuals exit self-isolation. In contrast, noticeable reductions with no such side effect can be observed for $\Gamma \geq 0.5$. On the other hand, a small value of $\tau_t = 0.01$ seems unable to produce a positive outcome. In real life, the latter scenario would occur if the tracing programme was very slow, missing too many contacts as a result, or if the digital contacts application failed to promptly notify many of its active users. Another noteworthy occurrence in Fig 7 is the bimodality of some of the curves. This effect has been previously noted for larger tracing rates and overlaps [7], being a rare artefact of fast incidence reductions that cannot be sustained by a $\tau_t < \beta$ any further.

Aside from outlining the effects of different testing strategies and tracing network overlaps, this experiment also hints at which parts of $\tau_r$'s and $\tau_t$'s parameter spaces are more relevant for exploration. To aid our search, we plot heatmaps of these parameter's achieved peak suppression for different levels of overlap (see Fig 8), and observe, as a result, that significant outcomes (i.e. distinguishable from simulation noise for $\Gamma \approx 0.5$ and beyond) are obtained when $\tau_r \geq 0.1$ and $\tau_t \geq 0.04$, while values $\geq 0.1$ should fall within the "adequate" region of a large spectrum of $\Gamma$ values.

## 3.3 Effects of average degree and app uptake

Further, we analyse the impact of the application uptake in scenarios with different average degrees (i.e. $K \in 10, 20$), and more appropriate testing and tracing strategies—i.e. $\tau_t, \tau_r \in \{0.05, 0.1, 0.2, 0.5\}$. For this trial, we set $N = 1000$, the asymptomatic probability to $p_a = 0.2$, and the non-compliance rate to $\eta = 0.001$ (with automatic isolation exit after 14 days), selecting a single $I_p$ node as the infection seed. The results are averaged over 750 simulations to reduce the variance induced by the smaller $N$.

Fig 9 shows the peak suppression achieved by each strategy given a specific adoption level. For $\tau_t = 0.05$, uptakes $r \leq 0.5$ generally give results within the noise region. Improving the contact tracing rate, however, leads to a noticeable decrease of this maximal point, even at smaller adoption levels. This is particularly true in the larger average degree case. Interestingly, deploying a wider-scale testing programme alone ($\tau_r = 0.5$) seems to lead to a considerable spread reduction which makes contact tracing less beneficial at achievable uptakes (even entirely profitless in the $K = 10$ situation).

Our findings suggest that a testing rate of $\tau_r = 0.1$ remains suitable in conjunction with contacts isolation not only for the previous experiment with $N = 10000$, but also in these smaller scale scenarios featuring different average degrees. Consequently, we decided to examine

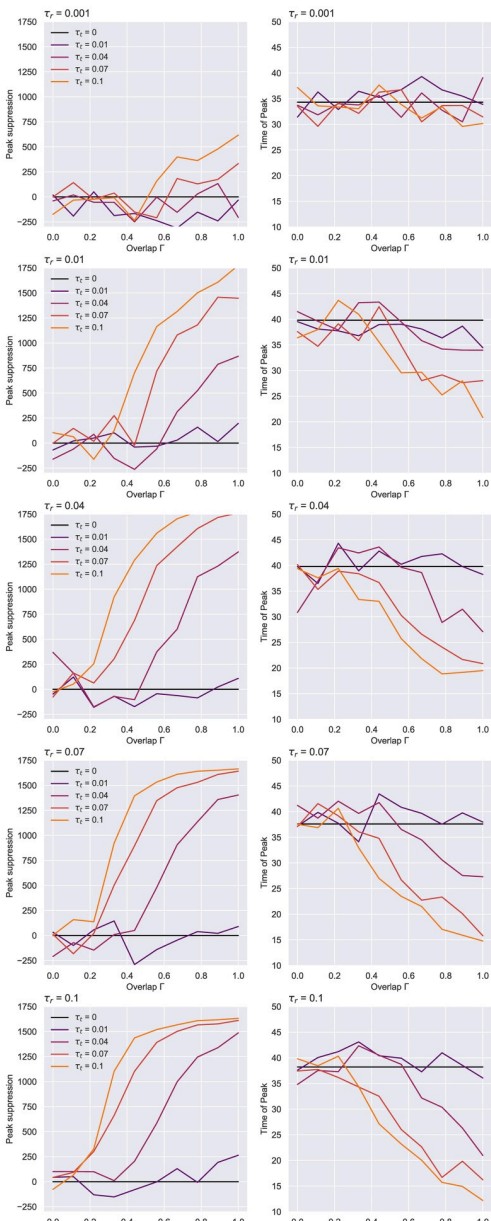

**Fig 6. ER network—Peak suppression (left) and the time of peak (right) at various tracing network overlaps.**
Values are averaged over 105 runs, representing results for $N = 10000$, $K = 10$, $p_a = 0.2$. The suppression is calculated
by subtracting the average maximal infected point given by each parameter configuration from the average point
obtained with no contact tracing ($\tau_t = 0$). Apart from $\tau_r = 0.001$ and $\tau_t = 0.01$ which produce inconclusive results that
we regard as noise, the effectiveness of an epidemic containment strategy expectedly scales with the testing and the
tracing rates.

further the effect of such a testing regime on the evolution of the spread (Fig 10), the number
of total deaths (Fig 11) and hospitalizations (S1 Fig in S1 File) for $K = 20$. The first chart below
illustrates how the epidemic curves significantly "flatten" for uptakes $r \geq 0.4$, the effect being
more apparent as the contact tracing rate increases. The second diagram puts these results into
perspective by showing that the number of deaths can be reduced even with lower uptakes,

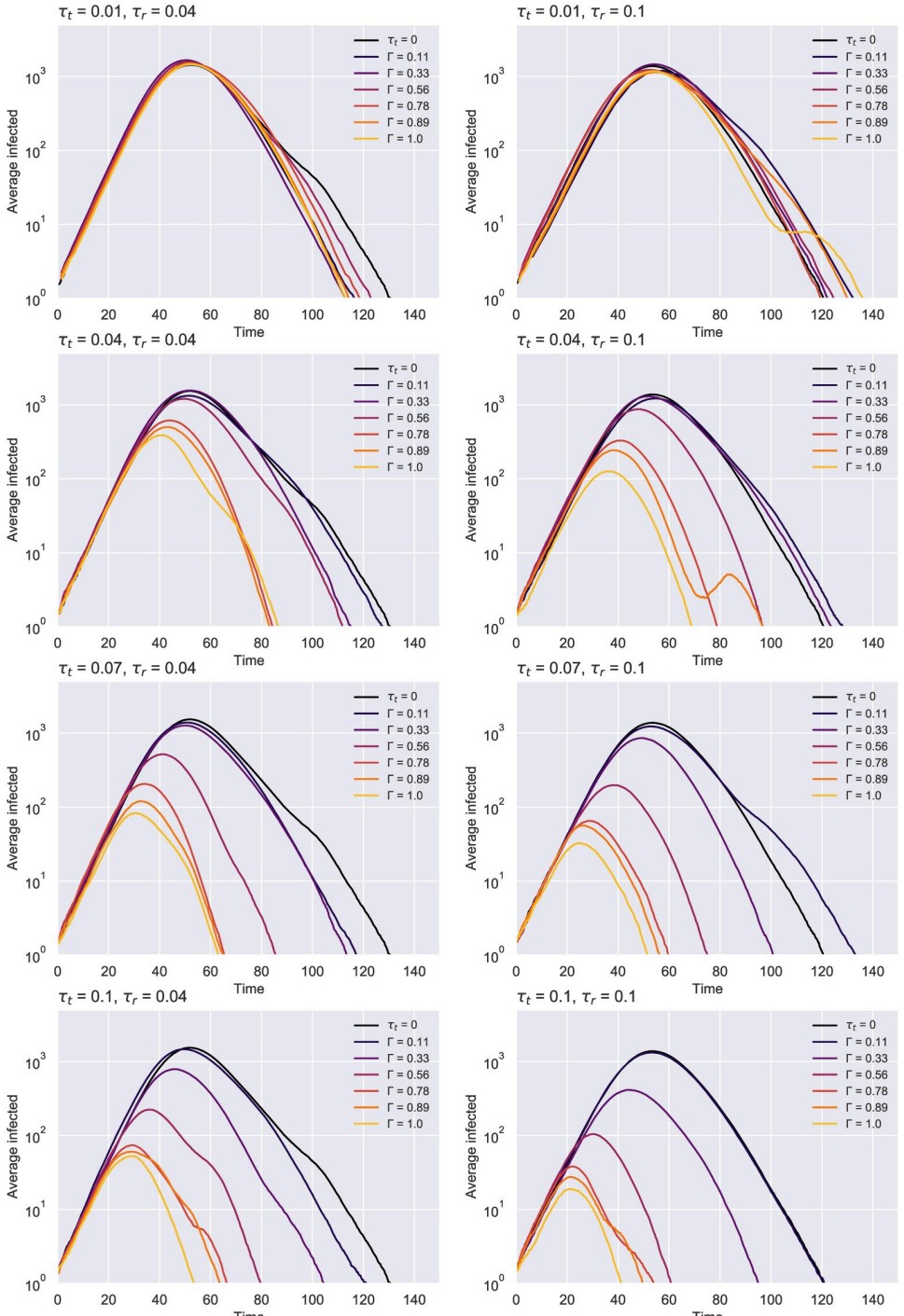

**Fig 7. ER network—Epidemic evolution over time given a less efficient (left) and a more effective (right) testing regime.** Results averaged over 105 simulations, obtained for $N = 10000$, $K = 10$, $p_a = 0.2$. As the contact tracing rate increases, the accuracy of the network given by $\Gamma$ becomes more important for "flattening" the curves. The case with no contact tracing ($\tau_t = 0$) is colored in black.

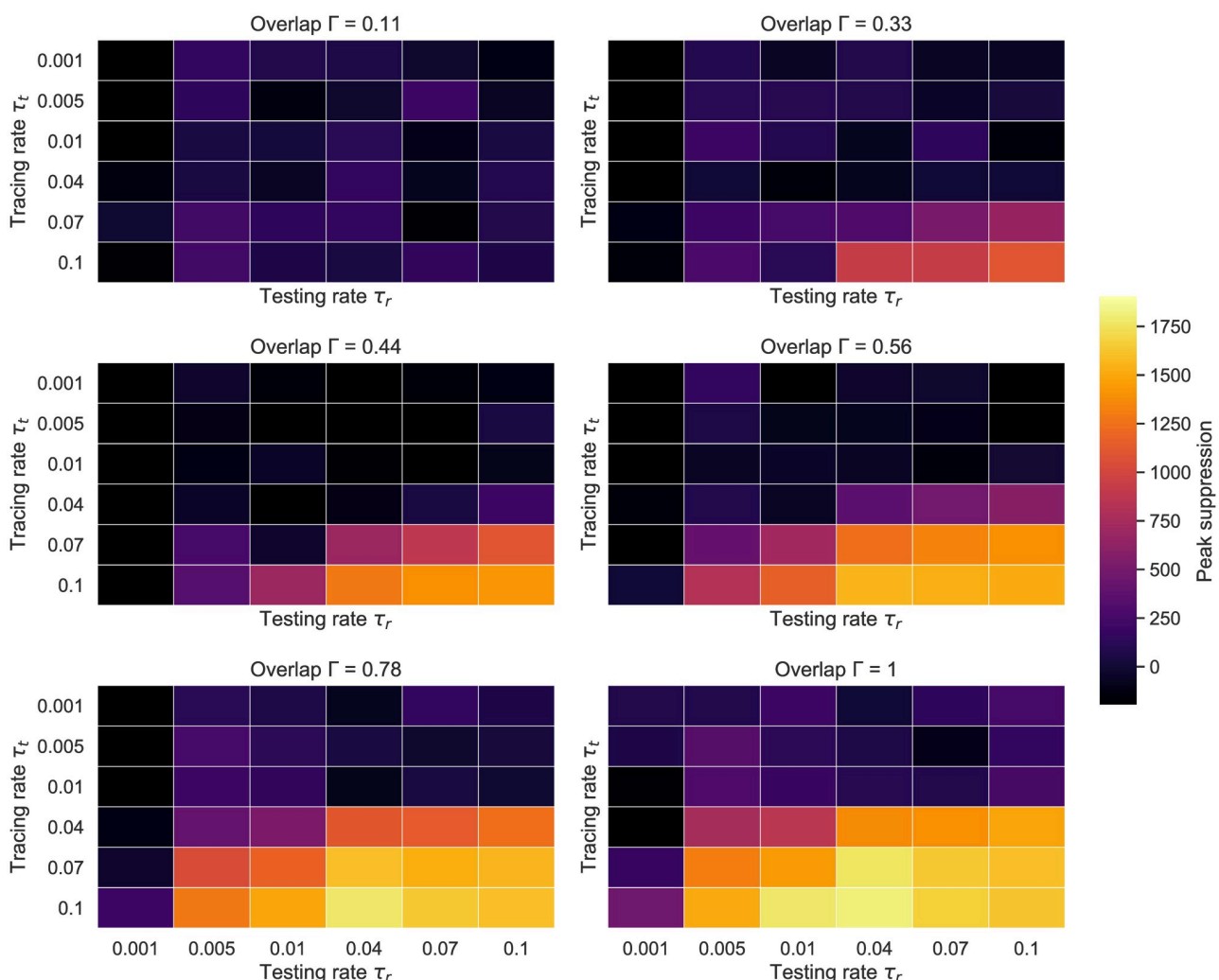

**Fig 8. ER network—Heatmaps of achieved peak suppression for different testing and tracing rates.** $N = 10000$, $K = 10$, $p_a = 0.2$. Averaged over 105 runs.

while at the other end of the spectrum many simulations have ended with notably fewer deceased (even none for an effective tracing $\tau_t \geq 0.2$).

Interestingly, these figures indicate that a higher uptake does not always guarantee a better epidemic outcome (e.g. $r = 1$ ends up with a higher peak than $r = 0.8$ in the case of $\tau_r = 0.1$ and $\tau_t \geq 0.2$). This is a direct consequence of isolating too many susceptibles early on in the outbreak (scenario similar to a partial lockdown), making their eventual self-isolation exit an unpredictable impact factor for the transmission chain.

### 3.4 Combining digital tracing with an imperfect manual tracing process

In this section, we study a more realistic scenario in which digital solutions complement an inherently imperfect interview-based tracing system. To that end, a *triad network* topology is employed, with digital tracing happening at a rate of $\frac{1}{\tau_t}$ days on average, over one subgraph given by the uptake $r \in \{0.1, 0.25, 0.4, 0.55, 0.7, 0.85, 1\}$, while the manual process gets carried at a slower pace of $2 + \frac{1}{\tau_t}$ days on average, over a third network view whose edges have been

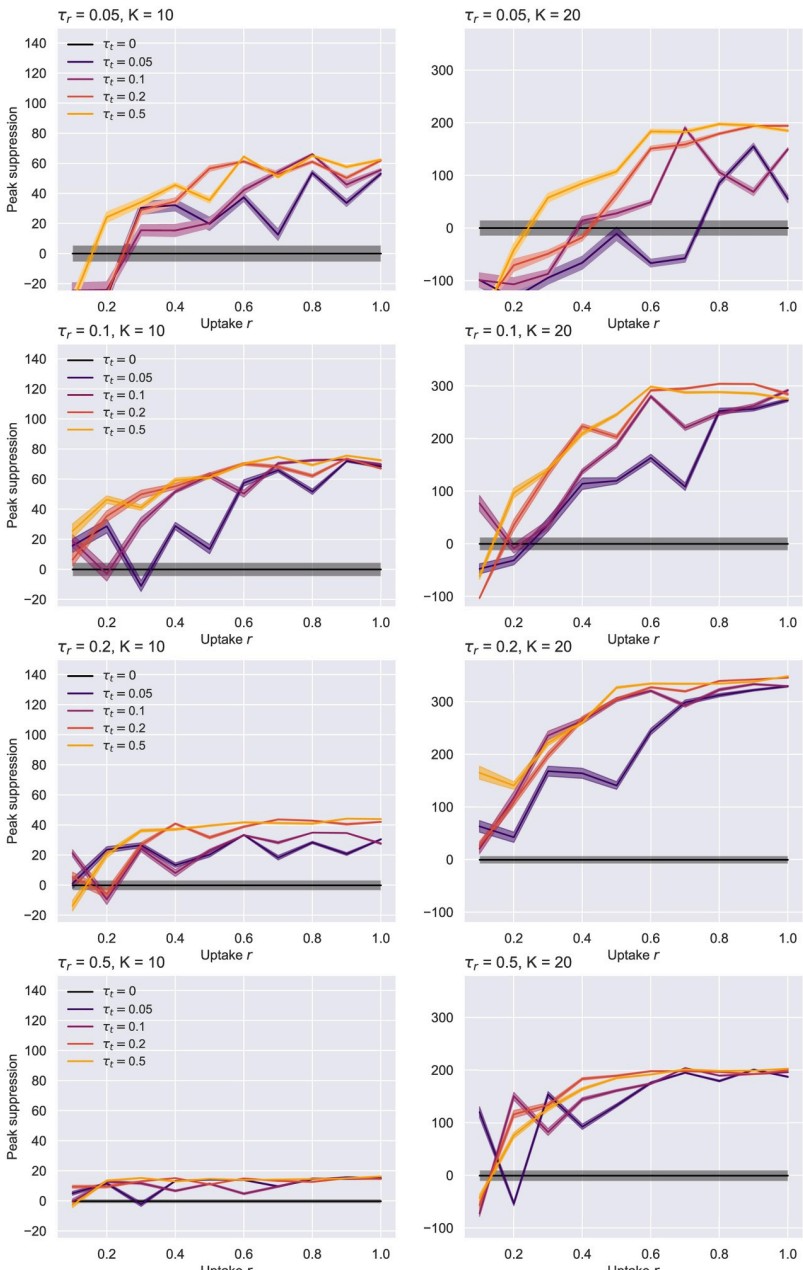

**Fig 9. ER network—Uptake rate $r$ against peak suppression.** Suppression is difference in peak to no tracing, i.e. $\tau_t = 0$. $N = 1000$, $p_a = 0.2$, $\eta = .001$. $K = 10$ given on the left, $K = 20$ on the right. The case with no contact tracing ($\tau_t = 0$) is colored in black. All lines were plotted with the 95% confidence intervals resulted from 750 runs.

randomly removed according to the degree of overlap $\Gamma \in \{0.1, 0.25, 0.4, 0.55, 0.7, 0.85, 1\}$. For the purpose of this experiment, we make use of a more representative graph structure for the SARS-CoV-2 transmission based on the Holme and Kim (HK) model [11], which features both a SF degree distribution and a SW clustering coefficient. The network parameters chosen here are: $N = 1000$, $m = 10$ (number of random edges to add for each new node; this replaces $K$ in Eq 1 for calculating $N_{ute}$) and $p_\triangle = 0.2$ (probability of making a triangle after adding a random edge). To avoid runs in which the epidemic gets quickly contained by chance, the

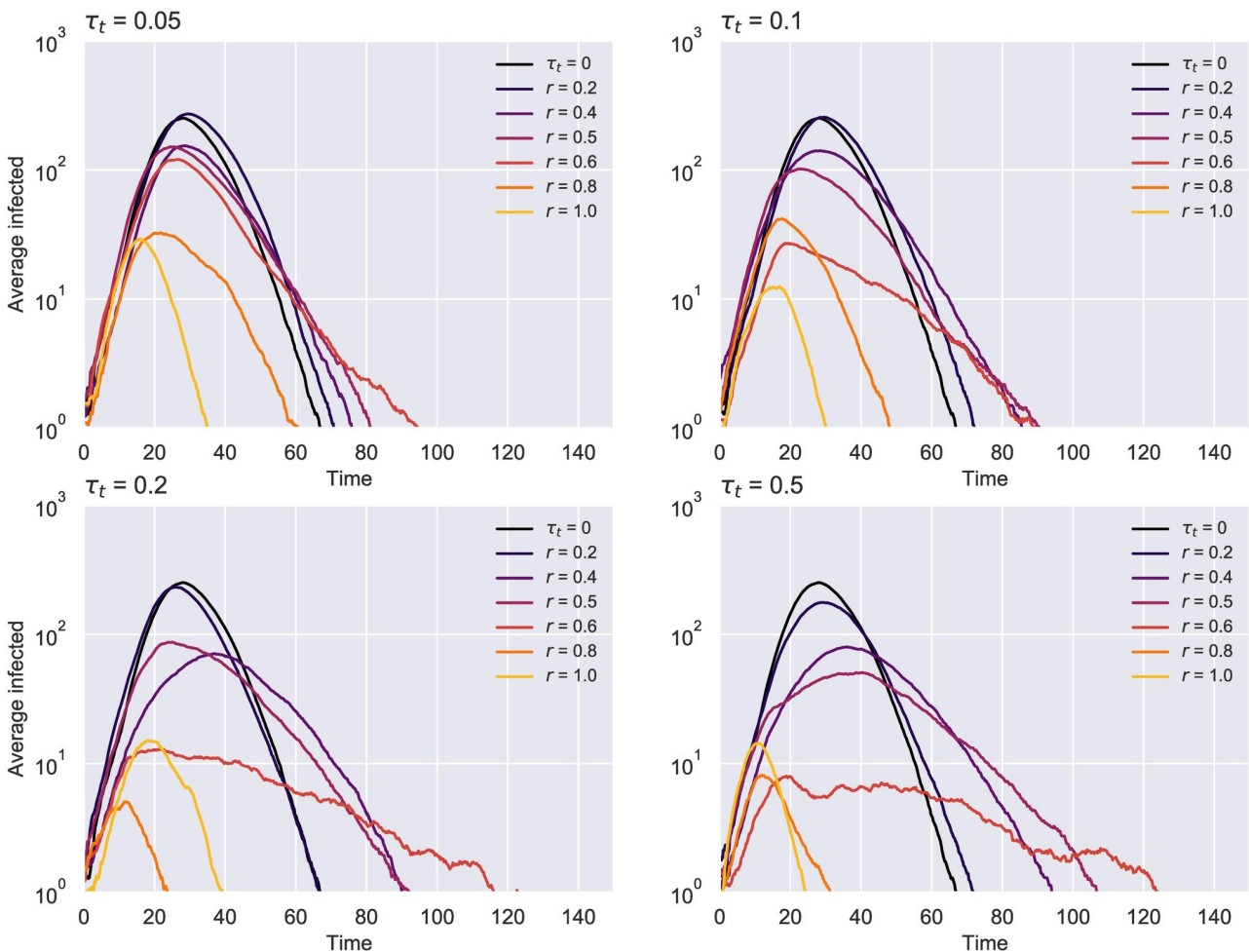

**Fig 10. ER network—Epidemic evolution over time for $\tau_r = 0.1$ $N = 1000$ and $K = 20$.** Results averaged over 750 runs. The case with no tracing ($\tau_t = 0$) is colored in black.

simulation starts with 10% of the nodes in the $I_p$ state—$c(t_0) = 10\%$ of $N$. The other parameters remain unchanged from the previous section, including the number of total runs.

The first aspect to notice in Figs 12 and 13 is that all curves remain monotonic with respect to $r$, while the dissimilarities between different $\tau_t$ contact rates become more apparent than what could be observed in the preceding experiment. This is a direct consequence of the increased number of infected people selected for time $t_0$, which prevents simulations from averaging over too many early-stopped runs. Considering the scale this pandemic has reached and the unavoidable presence of a delay between the infection onset and the debut of tracing, scenarios such as this one are more likely to occur, and therefore of a greater interest [17, 70].

Fig 12 shows the degree of peak suppression achieved by utilizing digital and manual tracing solutions when compared to a scenario in which no contact tracing was performed. These results suggest that, as the efficacy of the interview-based process increases (i.e. less contacts get missed), lower and achievable application adoption rates (20–50%) are sufficient to effectively reduce the maximal point of the epidemic. When the tracers are eventually able to "see" the full network of contacts ($\Gamma = 1$), varying $r$ no longer impacts the spread significantly, as

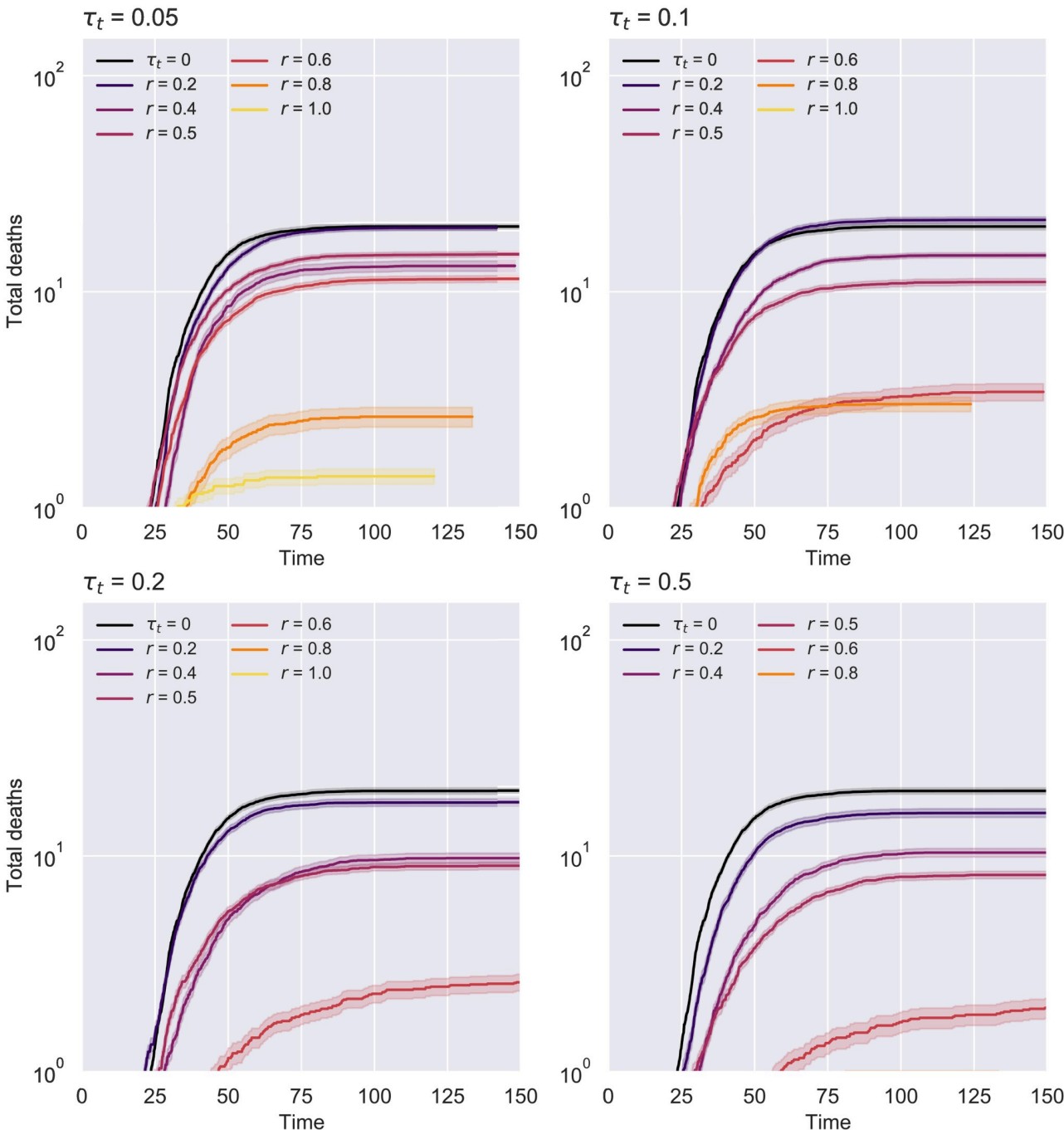

**Fig 11. ER network—Total deaths over time for $\tau_r$ = 0.1 $N$ = 1000 and $K$ = 20.** The 95% confidence intervals resulted from 750 runs are displayed around each line. The case with no contact tracing ($\tau_t$ = 0) is colored in black.

should be expected. In contrast, a very good testing regime ($\tau_r \geq 0.2$) can partially compensate for an inefficient manual tracing system ($\Gamma$ = 0.1) within the aforementioned uptake range.

Our estimate of $R$ = 3.20 for minimal interventions (i.e. $\tau_r$ = 0.05 and no tracing) during this scenario's first week falls within the confidence interval of the basic reproduction number $R_0 \in$ [3.09, 3.24] derived in Di Domenico et al. [24] by applying the next-generation approach

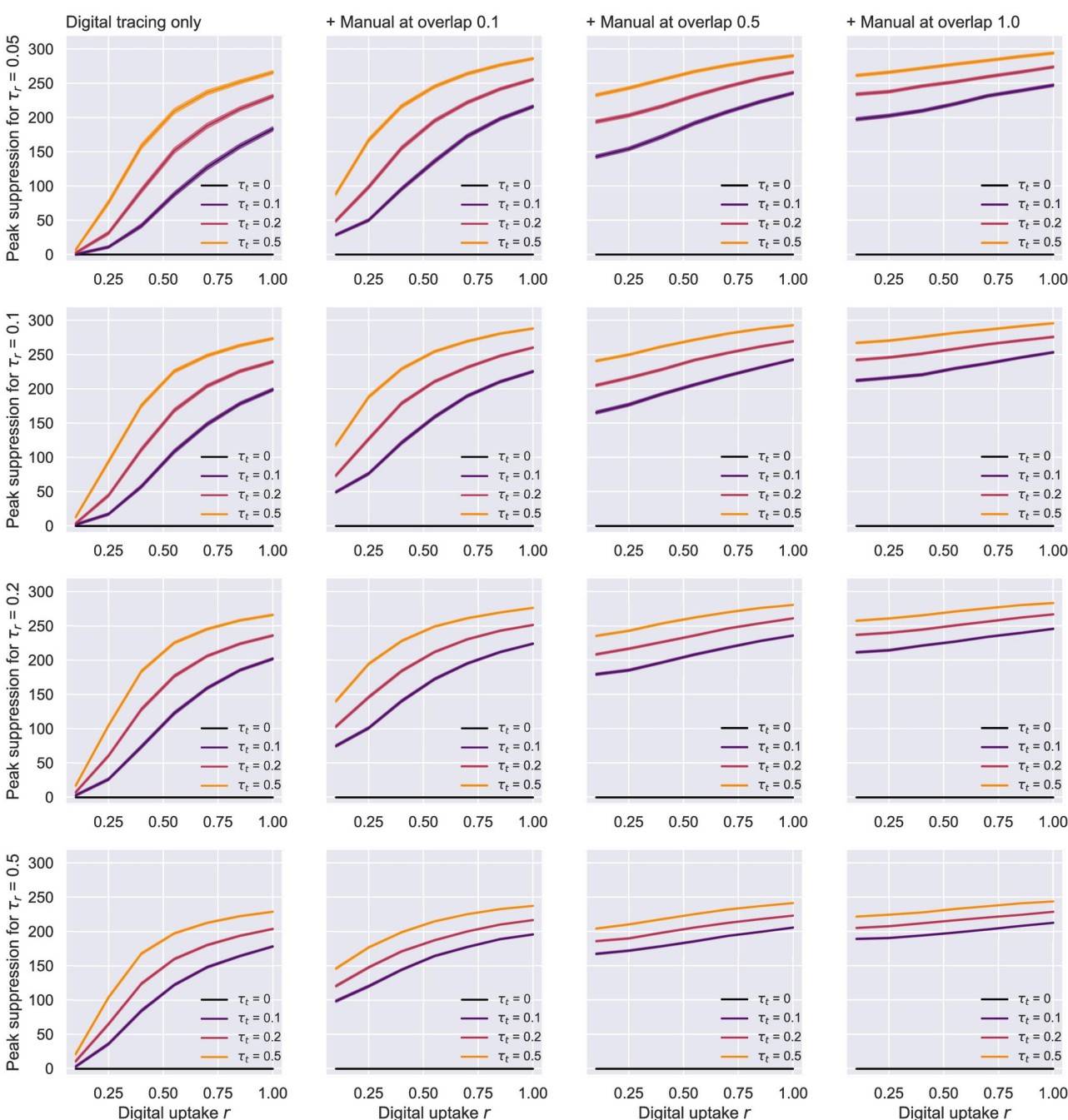

**Fig 12. HK network—Uptake rate *r* against peak suppression.** Suppression is difference in peak to no tracing, i.e. $\tau_t = 0$. The results here correspond to a Holme-Kim network with $N = 1000$, $m = 10$, $p_\triangle = 0.2$, $p_a = 0.2$, $\eta = .001$. On the left, we have a scenario in which only digital tracing was conducted, whereas the next 3 columns represent simulations with a combination of digital tracing on a second network, and manual tracing over a third network with various overlaps: 0.1, 0.55, 1. The 95% confidence intervals are displayed. The case with no tracing ($\tau_t = 0$) is colored in black.

[71] on a model fairly similar to ours. Fig 13 demonstrates that with good testing regimes ($\tau_r \geq 0.1$) and a reasonable manual tracing in place ($\Gamma \geq 0.5$), achievable uptake levels are enough to limit this $R$ to a value close to 1. In contrast, digital tracing alone fails to significantly reduce the spread unless both the testing and the adoption rates are very high. Similarly to what could

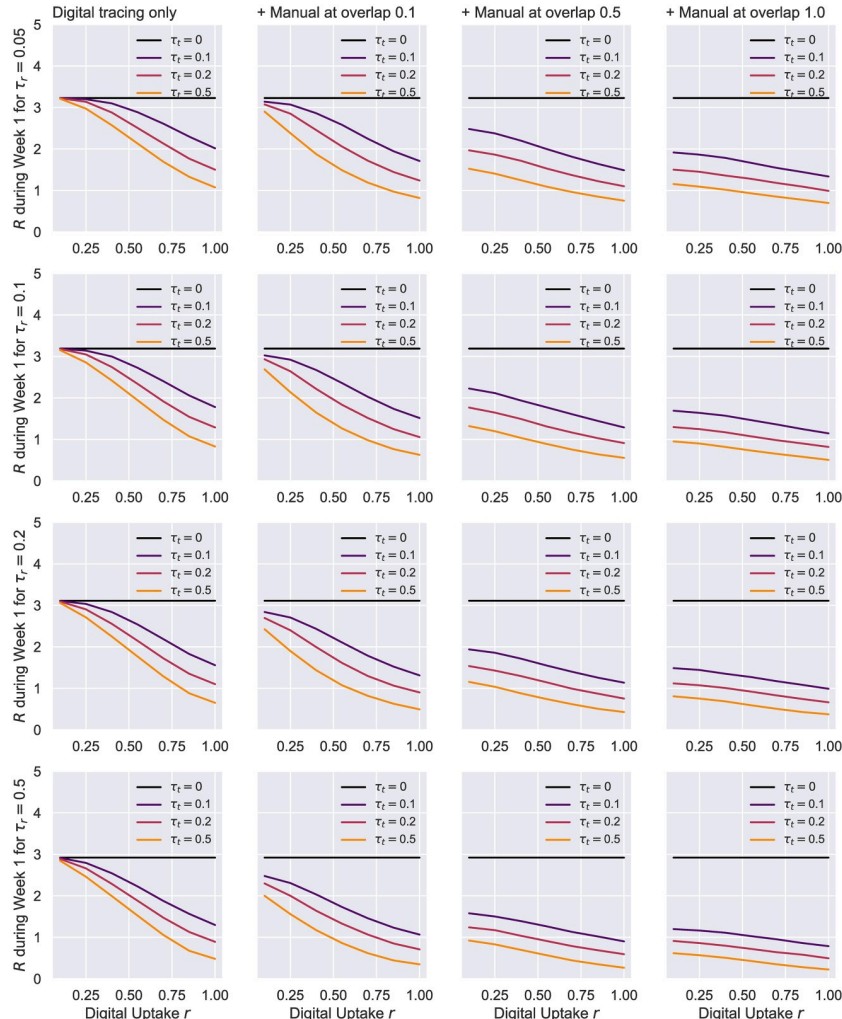

**Fig 13. HK network—Uptake rate *r* against the effective reproduction number *R*.** Suppression represents the difference to no tracing, i.e. $\tau_t = 0$. The results here correspond to a Holme-Kim network with $N = 1000$, $m = 10$, $p_\triangle = 0.2$, $p_a = 0.2$, $\eta = .001$. On the left, we have a scenario in which only digital tracing was conducted, whereas the next columns represent simulations with a combination of digital tracing on a second network, and manual tracing over a third network with various overlaps: 0.1, 0.55, 1. The case with no contact tracing ($\tau_t = 0$) is colored in black.

be observed in Fig 12, uptakes play a minor role in the infection proliferation if tracers are able to track the whole contact network eventually, yet this scenario is rather unlikely in real life. Interestingly, most of the other trends outlined in the peak suppression charts are faithfully mirrored by the evolution of *R* in the first week of the simulation. This reinforces the fact that efficient contact tracing in the early stages of an outbreak is essential for containing a virus like SARS-CoV-2 [72].

Even though peak suppression remains a good metric for assessing the benefits of public interventions, policy makers are more often interested in what combinations of these measures can quickly bring *R* to acceptable levels. In light of this, we plotted the contour lines of the *R* values produced by various degrees of interview-based network overlaps, testing and digital tracing adoption rates (see Fig 14, but also S6 Fig in S1 File). With an *estimated* uptake of around 40% in Finland and Ireland, 30% in the UK, or 27% in Germany and Norway at the time of writing [73], an effective testing regime ($\tau_r \geq 0.2$) coupled with an efficient contact

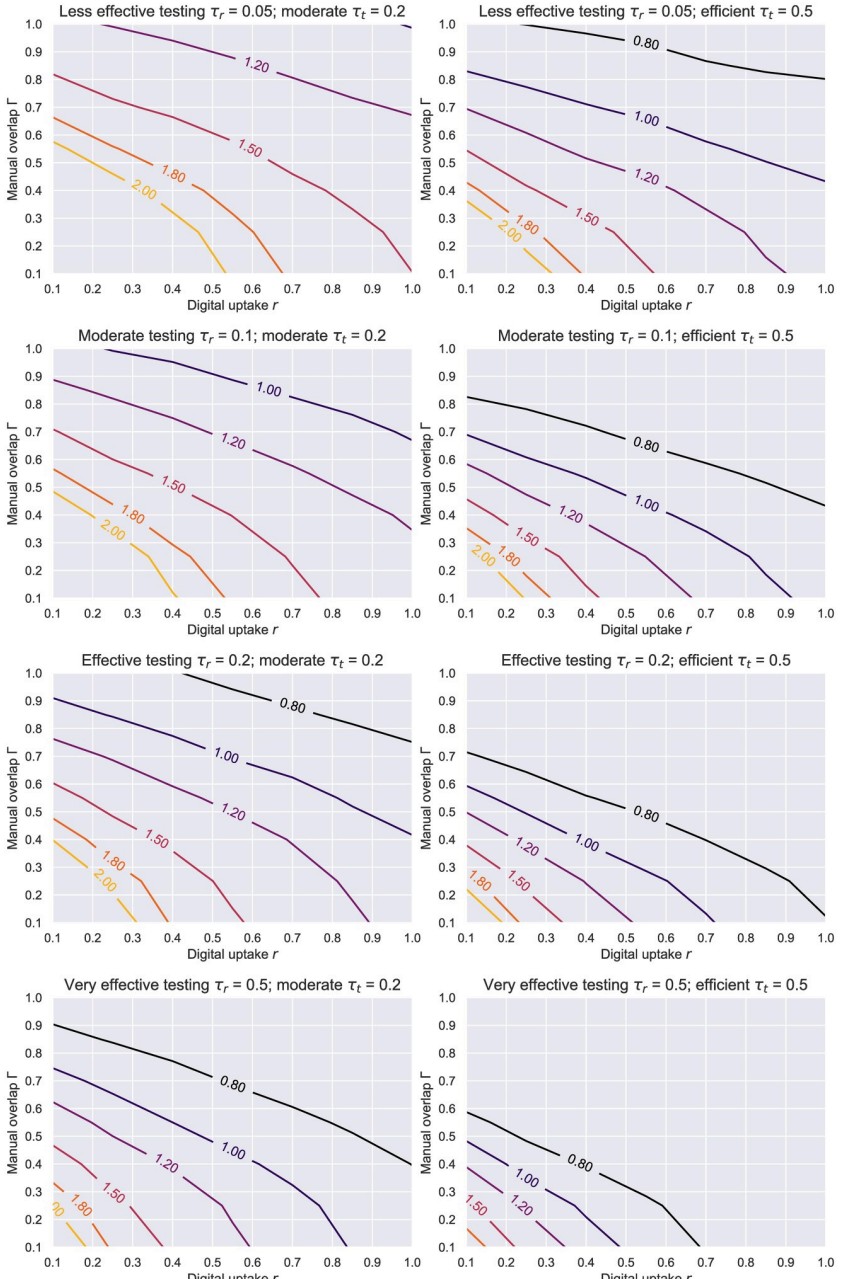

**Fig 14. HK network—Contour plots of $R$ based on the level of manual tracing overlap $\Gamma$ and digital tracing uptake $r$.** The results here are for a Holme-Kim network with $N = 1000$, $m = 10$, $p_\triangle = 0.2$, $p_a = 0.2$, $\eta = .001$. Each line represents a different testing level $\tau_r$, while the columns correspond to a moderate (left) and efficient (right) average level of tracing engagement and isolation compliance given by $\tau_t$.

tracing rate ($\tau_t = 0.5$) can drive $R$ below 1 even when tracers miss up to half the contacts ($\Gamma \geq 0.5$). Should this adoption improve to 50%, the aforementioned effect would be obtained with a testing rate half as good. In contrast, a moderate tracing rate only becomes effective if a large-scale testing programme gets deployed ($\tau_r = 0.5$) or bigger uptakes are achieved within a population ($r > 50\%$). We note the quoted uptakes were approximated using the total number

of application downloads, therefore due diligence should be exercised when interpreting these statistics because new downloads do not always convert to new active users.

## 3.5 Contact tracing efficiency in a real social network

Lastly, we evaluate the ability of digital tracing to curb an epidemic simulated over a real social network, in the presence or the absence of manual contact tracing. In this scenario, both the population size $N$ and the average degree $K$ are data-driven, with the latter also changing dynamically ($N = 74$, $K_{t_0} = 5.62$ at time $t_0$). Given that the network represents a tightly-knit community (static average degree $K_{static} > 60$), we investigate a broader range of testing and tracing rates: $\tau_r \in \{0.1, 0.2, 0.5, 1, 1.5\}$, $\tau_t \in \{0.1, 0.2, 0.5, 1, 1.5, 2\}$. The uptakes $r$, the overlaps $\Gamma$, and the initial incidence $c(t_0)$ are left unchanged from the last passage, while the relative delay between digital and manual tracing is kept at 2 days on average. The probability of becoming an asymptomatic case following exposure is fixed at $p_a = 0.2$ for the purpose of our initial discussion, but a comparison to the case in which $p_a = 0.5$ can be consulted at the end of this section.

The first thing to note about both Figs 15 and 16 is that each presents qualitatively similar trends to their counterpart figures from the previous experiment (i.e Figs 12 and 13, respectively). Namely, the better the testing and tracing rates are, the higher the benefit. At the same time, lower uptakes, in conjunction with an adequate overlap $\Gamma \geq 0.5$, consistently achieve significant peak reductions, driving the $R$ estimate of the first week below 1.5, even when the testing rate is smaller than 0.5. In contrast, with higher uptake values, the degree of overlap becomes less relevant for the epidemic outcome. Interestingly, the benefits of increasing the tracing effort $\tau_t$ beyond the value of 1 remain minimal across parameter configurations, and therefore we restrict further analyses to the range [0.1, 1].

Fig 17 presents the 2D contours of the estimated $R$ value (during the initial 7 days of the simulation) for the whole range of parameters. When comparing these results to Fig 14, we can see that a significantly faster testing strategy would be needed in this case to swiftly contain the epidemic and force $R < 1$. This is a consequence of dealing with an outbreak in such a lively and highly-interactive community, where the virus spreads too rapidly to afford testing at a lower rate than 0.5 (or even 1 in some cases) if the objective is to keep $R$ subunitary. Similarly, an efficient $\tau_t \geq 0.5$ is needed for achievable uptakes to attain (or be close to) the aforementioned goal. Where limited public health resources are available, locking down or restricting the movements within such hubs is therefore recommendable.

Finally, we investigate whether the efficacy of tracing appreciably changes when different $p_a$ values are considered. An asymptomatic node is assumed to be less infectious—$r^I = 0.5$, but also less likely to get tested positive—$r^T = 0.8$, so the epidemic dynamics should significantly differ when varying this probability. Remarkably, however, we observe the benefits of contact tracing do not fluctuate across the two studied values in the majority of the scenarios under scrutiny (see Figs 18 and 19). As shown in Fig 19, the most apparent differences in $R$ were recorded when less accurate tracing networks ($\Gamma$, $r < 0.5$) and less effective testing rates ($\tau_r < 0.5$) were employed. That is to say a suboptimal "test and trace" policy leads to more people getting infected when $p_a = 0.2$, yet this higher rate of infectiousness can be offset by the smaller likelihood of nodes testing positive in the $p_a = 0.5$ scenario, ultimately leading to minimal dissimilarities for the more adequate policies.

## 4 Conclusions and future work

This paper demonstrated how a novel methodology for modelling the effects of different "test and trace" strategies can be applied to study the transmission dynamics of a complex viral

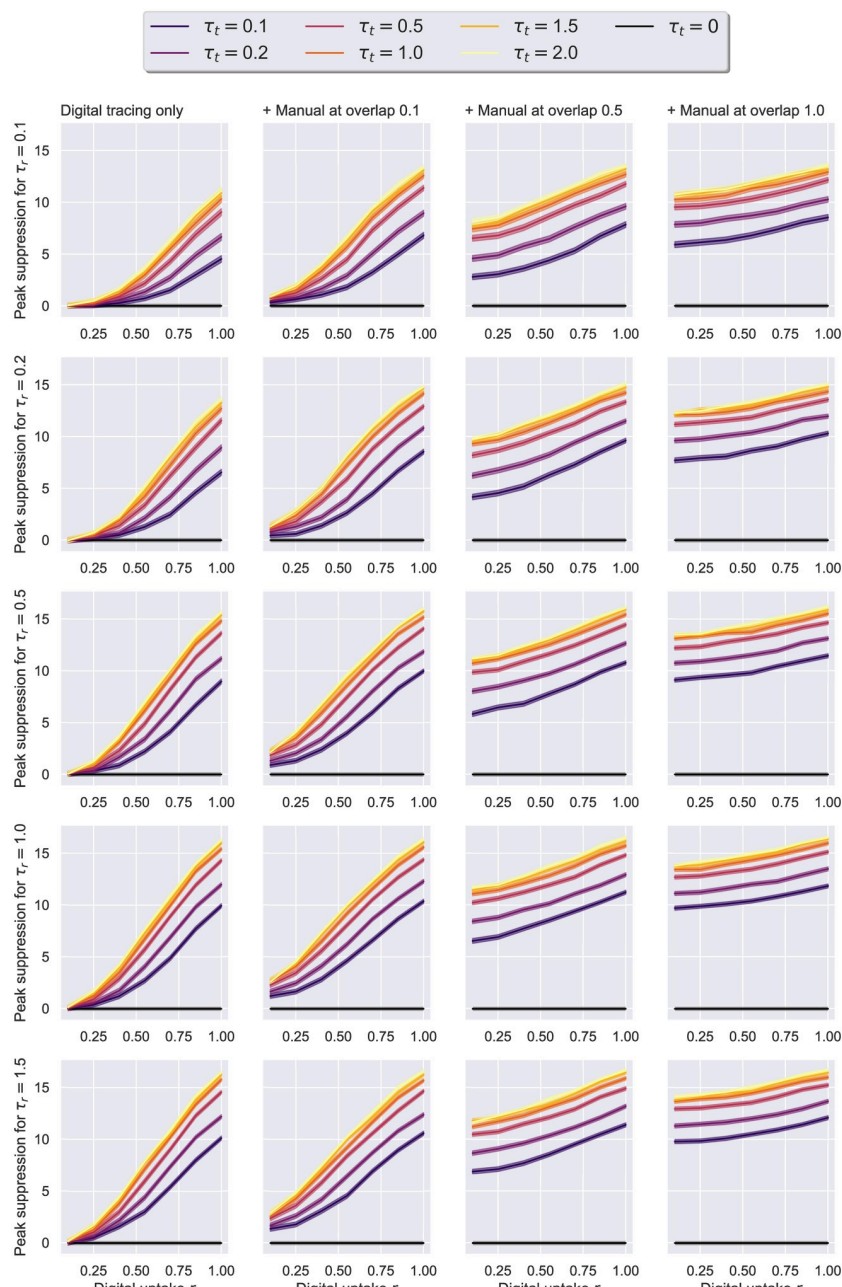

**Fig 15. Social evolution—Uptake rate _r_ against peak suppression.** Suppression is difference in peak to no tracing, i.e. $\tau_t = 0$. The results here correspond to the real Social Evolution network, dynamic over the studied period of 31 weeks, $p_a = 0.2$, $\eta = .001$. On the left, we have a scenario in which only digital tracing was conducted, whereas the next 3 columns represent simulations with a combination of digital tracing on a second network, and manual tracing over a third network with various overlaps: 0.1, 0.55, 1. The 95% confidence intervals are displayed. The case with no tracing ($\tau_t = 0$) is colored in black.

epidemic, such as COVID-19. Following a comprehensive analysis of the model's parameters, the procedures described here can be utilised to predict how the SARS-CoV-2 virus would spread through those communities where some indication of the interview-based network overlap and/or the digital tracing uptake exists. To facilitate such endeavors, we made our entire codebase open-source (refer to S1 File).

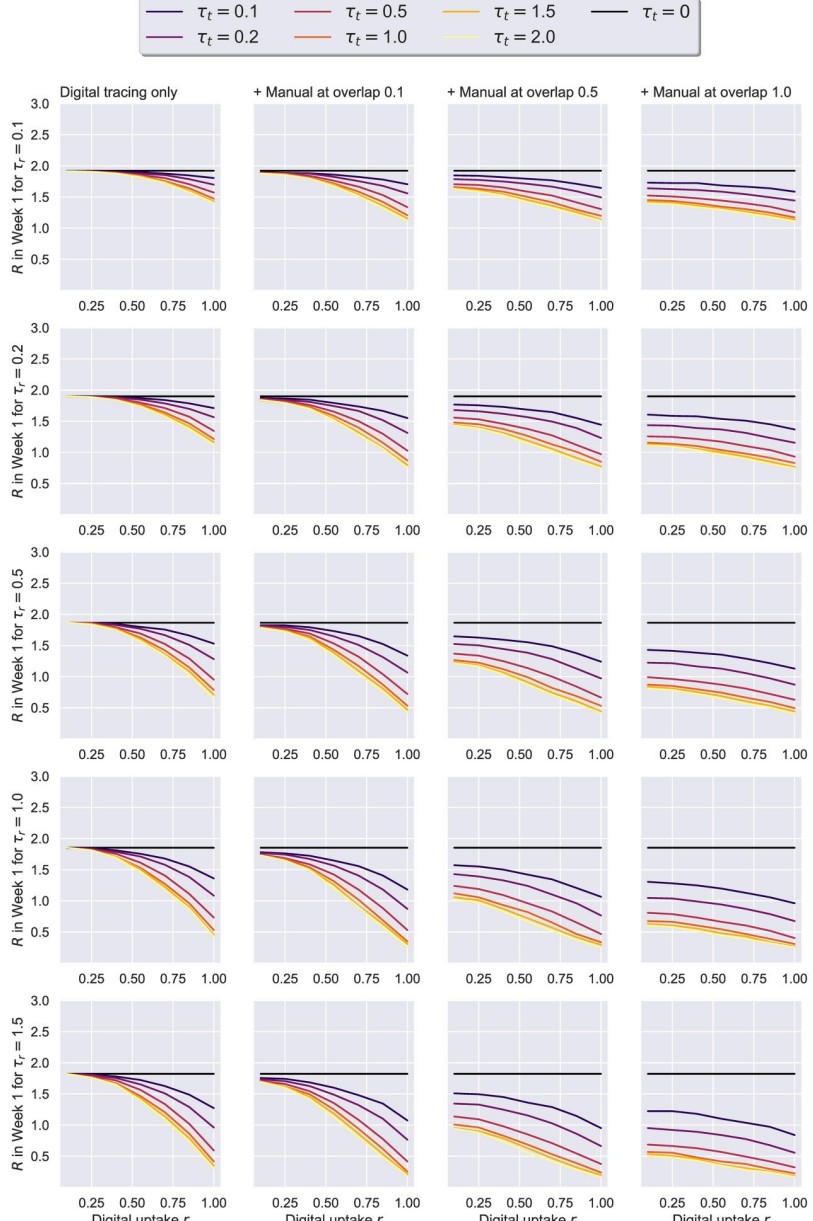

**Fig 16. Social evolution—Uptake rate *r* against the effective reproduction number *R*.** Suppression represents the difference to no tracing, i.e. $\tau_t = 0$. The results here correspond to the data-driven Social Evolution network, $p_a = 0.2$, $\eta = .001$. On the left, we have a scenario in which only digital tracing was conducted, whereas the next columns represent simulations with a combination of digital tracing on a second network, and manual tracing over a third network with various overlaps: 0.1, 0.55, 1. The case with no contact tracing ($\tau_t = 0$) is colored in black.

The approach we propose can address from a modelling perspective four of the open questions formulated by Anglemyer et al. in their Cochrane Review [74]: the combined effects of digital and manual tracing can be studied via the triad network topology, populations with poor access to the internet may be factored in by the degree of overlap Γ, individuals that have privacy concerns or accessibility issues can be represented in the system via the application adoption rate *r*, while the ethical and economical repercussions of balancing false positives and false negatives of tracing can be assessed through the statistics our simulations readily capture

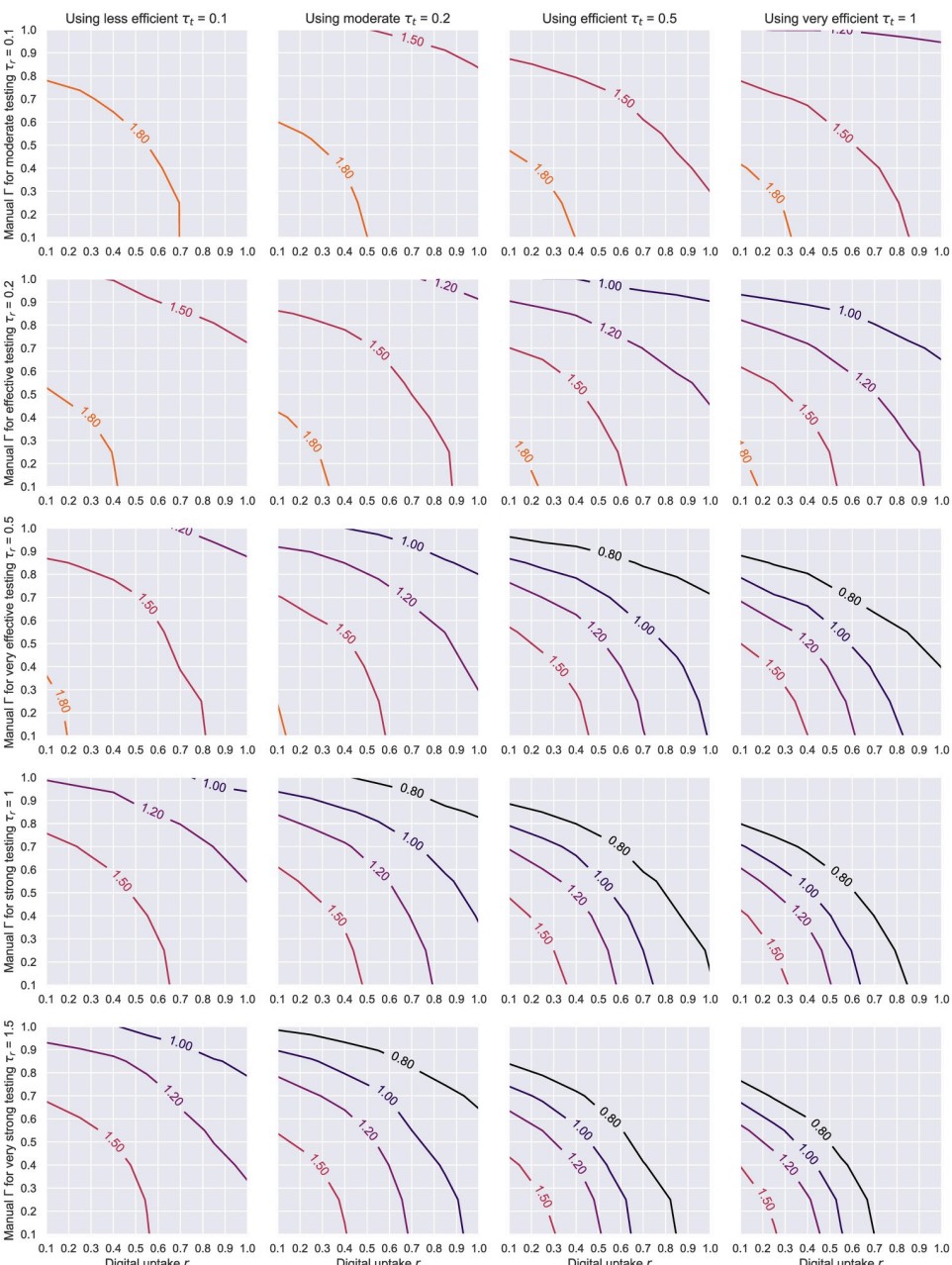

**Fig 17. Social evolution—Contour plots of $R$ based on the level of manual tracing overlap $\Gamma$ and digital tracing uptake $r$.** The results here correspond to the real Social Evolution network, dynamic over the studied period of 31 weeks, $p_a = 0.2$, $\eta = .001$. Each line represents a different testing level $\tau_r$, while the columns showcase a less efficient (far left), a moderate (center-left), an efficient (center-right) and a very efficient (far right) average level of tracing engagement and isolation compliance given by $\tau_t$.

(for more details, consult S2 and S3 Figs in S1 File). Consequently, the model we put forward is already powerful enough to answer a large spectrum of research and policy-related questions.

The simulations we conducted show that digital tracing remains a viable solution for reducing the peak of an outbreak, as well as the effective reproduction number $R$, even when its

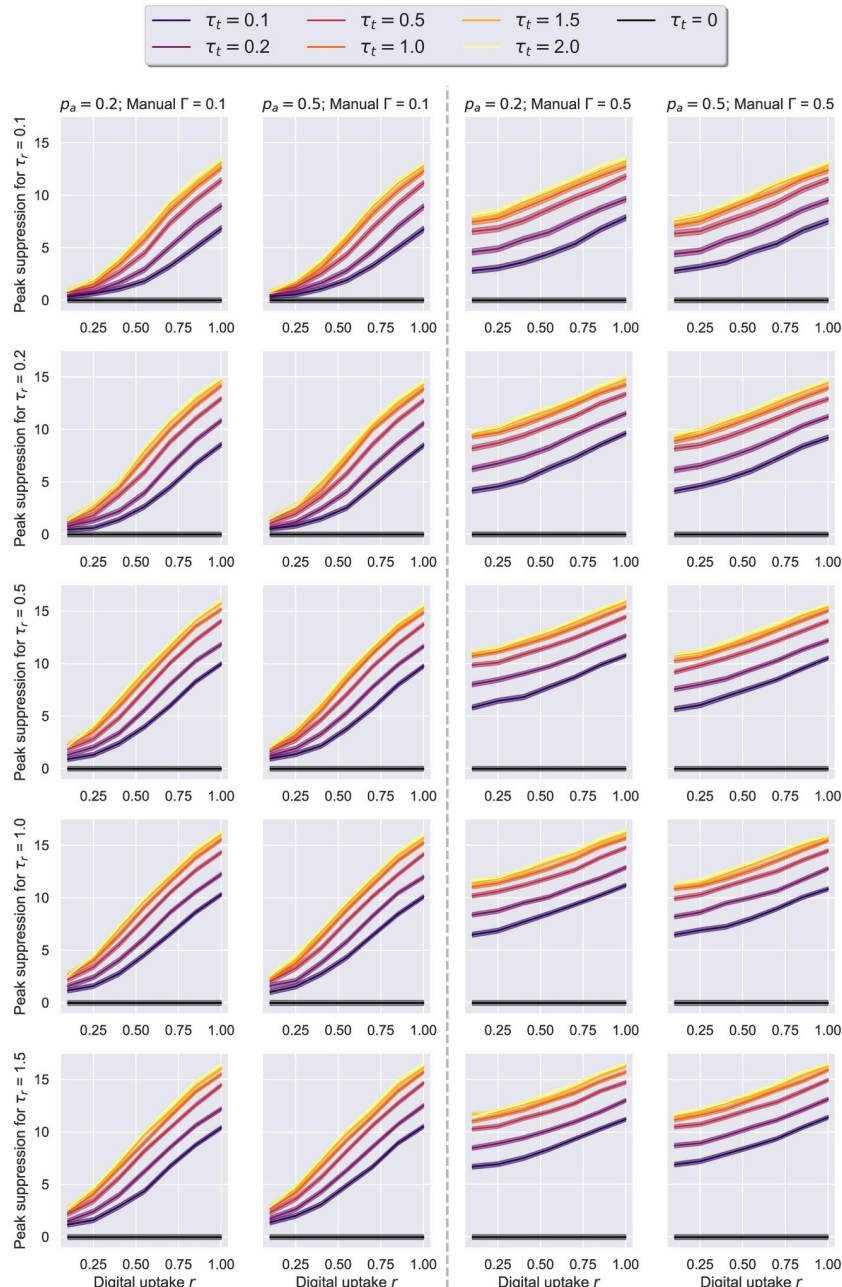

**Fig 18. Social evolution—Uptake rate $r$ against peak suppression, for different $p_a$ values.** Suppression is difference in peak to no tracing, i.e. $\tau_t = 0$. The results here correspond to the real Social Evolution network, dynamic over the studied period of 31 weeks, $\eta = .001$, and either $p_a = 0.2$ (on the left of each pair) or $p_a = 0.5$ (on the right of each pair). The left-quadrant pairs represent a triad network scenario with manual overlap $\Gamma = 0.1$, while the right quadrant showcases $\Gamma = 0.5$. The 95% confidence intervals are displayed. The case with no tracing ($\tau_t = 0$) is colored in black.

adoption levels are lower. At the same time, a less efficient interview-based process, which misses up to half the contacts, can still contain the spread if coupled with 30–40% application uptakes and large-scale testing regimes. For highly-connected communities, the latter condition becomes even more essential for swift containment. The peak reduction seems ubiquitously tied to how fast the tracing is conducted, as well as how impactful the public-health

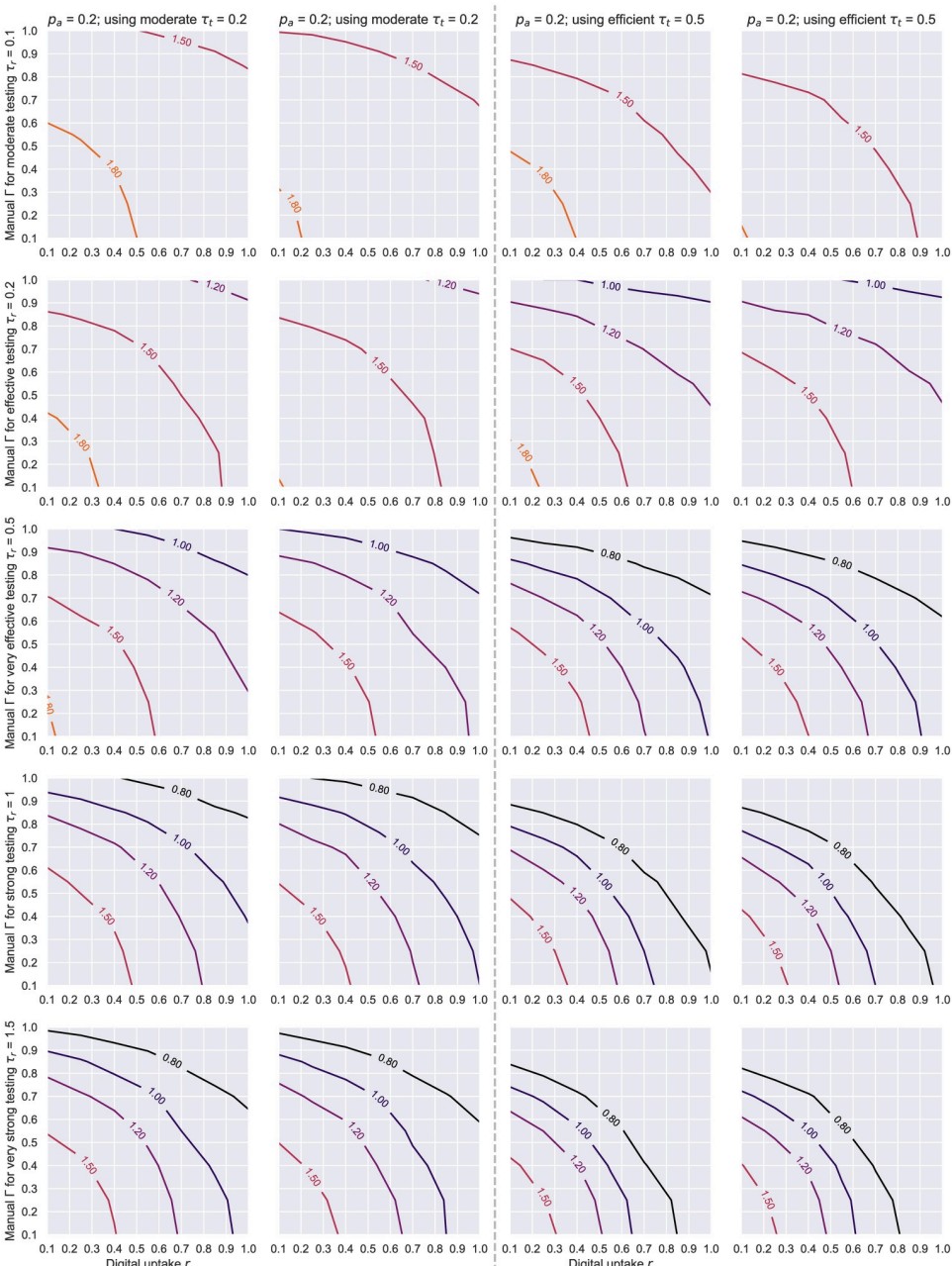

**Fig 19. Social evolution—Contour plots of *R* based on the level of manual tracing overlap Γ and digital tracing uptake *r*, for different $p_a$ values.** The results here correspond to the data-driven Social Evolution network when $\eta$ = .001, with each pair of charts describing $p_a$ = 0.2 on the left and $p_a$ = 0.5 on the right. Each line represents a different testing level $\tau_r$, while the columns showcase combinations of one $p_a$ value together with a level of tracing effort given by $\tau_t$.

messages are in making the involved communities more compliant with the self-isolation recommendations, as soon as more and more of each individual's contacts get traced and isolated (aspects encompassed in the $\tau_t$ rate).

We would like to emphasize that the parameter ranges under scrutiny in this study are by no means exhaustive. Therefore, we leave for future exploration studying the effects of

extensively varying the average degrees of the random networks, the non-isolation rates, the initial infections, or the variant-specific asymptomatic and hospitalization probabilities. Looking at such diverse scenarios would allow one to better estimate the shortfalls of contact tracing when different variants of concern are circulating, discover factors that may have introduced significant inefficiencies into the strategies adopted by many countries (e.g. higher non-compliance [75]), while also ensuring the variability induced by early-stopped simulations is curbed.

Next, we envision leveraging several mobility datasets in subsequent endeavors to infer a broader range of network structures, and derive time-dependent estimates of the transmission rate, as previously described in Liu et al. [60]. Other parameters in our model could be tailored to the epidemiological situation of different countries by fitting them to governmental data reporting on the number of COVID-19 deaths registered within each region of interest.

Finally, the random nature of testing and deriving static tracing views in this study may not provide the most realistic setup. Strategically targeting mass-testing campaigns to hubs or dynamically intensifying tracing efforts in highly-affected regions could significantly improve the outcome of an outbreak. To finely control the network dynamics in such an informed fashion, we expect future studies to utilize a combination of graph neural networks and gradient-based reinforcement learning techniques, possibly leveraging the setup recently proposed by Meirom et al. for prioritizing the viral testing allocation [45].

## Supporting information

**S1 File. Supporting material.** Contains a link to the repository that maintains our open-source model, further discussions on other epidemic statistics we captured, and more charts illustrating the effects induced by varying the contact tracing parameters.
(PDF)

## Acknowledgments

We would like to extend our gratitude to Professor Niranjan Mahesan for his novel ideas, as well as his continuous effort and support throughout this project. We also acknowledge the use of the IRIDIS High Performance Computing Facility, and its support services at the University of Southampton, in the completion of this work.

## Author Contributions

**Conceptualization:** Andrei C. Rusu, Rémi Emonet, Katayoun Farrahi.

**Formal analysis:** Andrei C. Rusu.

**Investigation:** Andrei C. Rusu.

**Methodology:** Andrei C. Rusu, Rémi Emonet, Katayoun Farrahi.

**Project administration:** Katayoun Farrahi.

**Software:** Andrei C. Rusu, Rémi Emonet.

**Supervision:** Katayoun Farrahi.

**Visualization:** Andrei C. Rusu.

**Writing – original draft:** Andrei C. Rusu.

**Writing – review & editing:** Andrei C. Rusu, Rémi Emonet, Katayoun Farrahi.

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
