## [Decision Letter · Decision Letter 0]

20 May 2021

PONE-D-21-07031

Modelling digital and manual contact tracing for COVID-19. Are low uptakes and missed contacts deal-breakers?

PLOS ONE

Dear Dr. Rusu,

Thank you for submitting your manuscript to PLOS ONE. After careful consideration, we feel that it has merit but does not fully meet PLOS ONE’s publication criteria as it currently stands. Therefore, we invite you to submit a revised version of the manuscript that addresses the points raised during the review process.

We look forward to receiving your revised manuscript.

Kind regards,

Ivan Kryven

Academic Editor

PLOS ONE

Journal Requirements:

Reviewers' comments:

Reviewer's Responses to Questions

**Comments to the Author**

1. Is the manuscript technically sound, and do the data support the conclusions?

Reviewer #1: Partly

Reviewer #2: Yes

2. Has the statistical analysis been performed appropriately and rigorously? 

Reviewer #1: Yes

Reviewer #2: N/A

3. Have the authors made all data underlying the findings in their manuscript fully available?

Reviewer #1: Yes

Reviewer #2: Yes

4. Is the manuscript presented in an intelligible fashion and written in standard English?

Reviewer #1: Yes

Reviewer #2: Yes

5. Review Comments to the Author

Reviewer #1: The authors present a model combining the spread of an epidemic with a mitigation strategy based on contact tracing and isolation. The model is a multi-site mean-field model combining equations for the evolution of compartment size and the use of an underlying network of contacts between agents. Exploring the range of parameters, they test several scenarios of tracing efficiency and their impact on outbreaks. They find that imperfect tracing can nonetheless have a sufficient impact to significantly lower the epidemic peak.

The topic is interesting and addresses the very actual challenge of quantifying the impact of realistic mitigating strategies on epidemic outbreaks. As such, the study is perfectly valid and presents interesting results that could inform health policies.

I have however some issues with the model and some claims the authors make:

- The compartmental model used for describing the epidemic process is much more complicated than the classic ones. It is understandable, as the authors aim at describing realistic outbreaks of SARS-COV-2, in particular with the use of the Ip (pre-contagious), Ia (asymptomatic) and Is (symptomatic) compartments. However, the separation between H (hospitalised), R (recovered) and D is not necessarily useful, as D will only be a fraction from H set by lambda_H-D, and all remaining agents in the H compartment end up in the R one.

- The mechanisms for generating the tracing networks are not described. How are the links chosen, when, and by which mechanism? Is the tracing connected to new infections or not? If not (as this is a mean-field approach), the model might have an intrinsic limitation due to the absence of correlation between the infection process and the network.

- On the same topic, at line 127 the authors say: "The tracing graphs are usually subsets of the first network." When is this not the case? From the model described in ref 21, I get that tracing might generate "false" contacts, but this is not said at any point in the manuscript.

- The authors validate most of their findings using Erdös-Rényi networks, claiming that "it tends to offer acceptable approximations most of the time" (l146). I must argue that it does not, and in fact the authors themselves then use a scale-free/high clustering model in section 3.4. Why not directly use this one, which is indeed more realistic? They also could use empirical datasets, which are easily found in the community.

- Another problem is the temporality of the spreading. Line 155, the authors state: "The time intervals between two state changes of the same kind are assumed to form an exponential distribution". This is very unrealistic. As shown in a wide range of empirical studies, human interactions exhibit bursty behaviour, with distributions of temporal properties having typical heavy-tails. It has been further shown that such properties condition strongly the spread of an epidemic (see for example Lambiotte et al EPJB 86 (2013) 320).

- I suggest the author define the measures they used to quantify the effect of tracing in the text. In particular, what is "peak suppression"? I would recommend a clearer measure, such as a normalised peak reduction (N_(no tracing) - N_tracing)/(N_(no tracing)). Same approach could be used for time of peak.

- l203 : The authors select "good" values for tau_r and tau_t, but it seems to me that there is a correlation between the threshold values of tau_r and tau_t. It would be interesting to see heatmaps combining both dependencies.

- I have a question about the results presented in Fig 5, top right quadrant. It seems that the larger the population, the smaller the maximum infected fraction of the population, with a limit going to 0 as N increases. This would indicate that the model does not generate large cascades of infections, while spreading processes —being analog to percolation— should always reach a non-zero fraction of the population when the infection parameter is above the epidemic threshold. If accurate, doesn't it indicate an intrinsic unrealistic property of the model? It could also be that the parameters chosen for the study in section 3.1 lead to a spreading under the epidemic threshold, but in that case the analysis of the variation induced by population size is irrelevant.

- The study would be much stronger if the model was explored and validated for a larger range of spreading parameters. Only two values for p_a and one for p_h are considered. I understand that the authors have used estimated values from the empirical studies, but since the model for interactions is not realistic, it might be that these values are not suitable to reproduce the desired spreading properties (see my previous comment).

- Similarly, only two values for K are considered.

- I understand that the model is complex and the parameter space is huge, but the current results merely rely on eye-balling "relevant" values from a minimal set of tries. Deriving quantitative, meaningful results from such an evaluation of "proper" values for the parameters seems too optimistic.

- Manual tracing and digital tracing seem to be a sort of redundant system, to increase the global tracing efficiency. Shouldn't the use of both be equivalent to having a single, higher value of either digital tracing or manual tracing?

- I would appreciate that the authors discuss two features from Fig 7:

1. How can contact tracing lead to situations in which the outbreak lasts longer?

2. What generates the bimodality of some curves?

Furthermore, I have some minor issues with the manuscript:

- I would advise for replacing ref 21 with "R. Huerta, L. Tsimring, Phys. Rev. E 66 (2002) 056 115". I am not an expert on multi-site mean-field models, and I found that the latter reference contains a much clearer description of the approach than the one given by the authors.

- l196 : please move the reference to Fig 7 at the relevant location ; the current one is a sentence about Fig 6.

Reviewer #2: The paper presents a novel (SEIR-type) compartmental model of epidemic spread that accounts for possible manual and/or digital contact tracing by describing the individuals’ state by a pair of variables, one that describes its epidemic state and a second that describes whether it is traced (and isolating) or not. The authors explore the properties of the model using simulations, and show on synthetic data that combined manual and digital contact tracing may be effective in curbing the simulated spread of SARS-COV2 even with either of the two tracing modalities is suboptimal.

The paper is generally technically sound and the writing is clear and understandable. Though, given the large number of different parameters and tests performed it is sometimes difficult for the reader to recall everything and follow the story. I recommend the manuscript for publication in PLOS ONE after the authors have addressed several, mostly technical, points and some optional suggestions that I think may help the reader when reading the paper.

The conclusions of the simulation study are weakened by the fact that they rely solely on simulated contact networks which lack many realistic features of real world contact patterns. I think the model in itself and the present study merit publication on their own, but it would be a big plus if the authors could motivate that the networks’ parameters are realistic and that the missing features do not change the conclusions, and/or perform simulations on empirical contact networks.

The model has a lot of different variables and parameters and it is difficult to keep track of them all. It would be very helpful to list all of them in a table, similar to Table 1. Namely: \\Gamma, K, r, t_rem. It might also helpful to list the different compartments (states) of the model.

In the same manner, it would make the manuscript easier to read if the names, or short definitions, were given for each of the parameters when their ranges considered are listed in each subsection of the Results, e.g., on lines 187 and 188 recall that \\Gamma is the degree of overlap, \\tau_t the contact tracing rate, etc.

Add values of all relevant parameters to figure captions. This will make the figures easier to read.

It is unintuitive to use S_i for general states and S for the susceptible state. I suggest the authors use another symbol to denote a general state, e.g. X_i.

The fact that the relative error is ~3% for networks of size N=1000 and ~1% for N=10000 is corresponds to a relative error of 1/\\sqrt{N}, consistent with central limit type arguments for a well mixed population. The result may not hold however in a structured population or near a critical point and thus cannot necessarily be extrapolated to general conditions.

The authors refer to some parameters of their model as “hyperparameters”. This term is generally reserved for parameters of an inference procedure, not a model, e.g., of the learning procedure in machine learning or of prior distributions in Bayesian inference.

The authors compare the uptake rate in the simulations to the digital tracing adoption rates reported for different countries. However, the adoption rates cited are typically calculated as numbers of download relative to the population, which is not the same as the fraction of population using the app correctly.

In Section 3.4, if I understood correctly, the overlap between edges in the digital contact tracing network and the true contact network is assumed to be perfect, while the uptake in the manual tracing network is assumed to be 100%. Both of these assumptions seem overly optimistic to me, e.g., the manual contact tracing system will saturate at a given prevalence as was seen in most western countries, while digital contacts are only proxies for real (possibly disease transmitting) contacts.

Please corroborate the claim (on lines 144-146) that the Erdos-Renyi graph model tends to offer acceptable approximations most of the time.

It seems like an error that symbols in Table 1 are in boldface.

The authors may want to discuss the implications of the crossover effects seen in Fig. 10, i.e., the final outbreak size can be larger for higher uptake.

Can the authors comment on how realistic the required values of \\Gamma and \\tau_t for combined tracing to be effective alone are? Have the values of these parameters been estimated empirically?

In Fig. 5 add description of what symbols and boxes and whiskers represent.

Line 74: Change “SIR” to “The SIR process”

Line 80: add abbreviation “(Inserm)”

line 121: “onto” → “on”

Lines 140-141: should “scales with” be replaced by “is proportional to”?

Line 205: Specify what is meant by “meaningful results”.

Line 212: “N” → “small N”.

Line 302: “encapsulated” → “model’s”

6. PLOS authors have the option to publish the peer review history of their article (what does this mean?). If published, this will include your full peer review and any attached files.

Reviewer #1: No

Reviewer #2: No

---

## [Author Response · Author response to Decision Letter 0]

6 Jul 2021

Replies to editor:

Following the editor’s comments, we amended the supporting information references to respect the format – i.e. S1 Fig instead of Fig S1. We also limited the use of italics to very important information only. Finally, we changed the formatting of the authors section to match the suggested styling.

Replies to reviewer #1:

1. The compartmental model used for describing the epidemic process is much more complicated than the classic ones. It is understandable, as the authors aim at describing realistic outbreaks of SARS-COV-2, in particular with the use of the Ip (pre-contagious), Ia (asymptomatic) and Is (symptomatic) compartments. However, the separation between H (hospitalised), R (recovered) and D is not necessarily useful, as D will only be a fraction from H set by lambda_H-D, and all remaining agents in the H compartment end up in the R one.

Our goal from the very beginning was to make simulations that could capture a broad range of effects stemmed from the viral spread, including the evolution of hospitalizations and deaths. We also wanted to show that, unlike previous works, our contact tracing approach can be used in conjunction with virtually any complex compartmental formulation (many of which have been proposed for COVID-19 given the inefficacy of SIR [1]). That being said, this modeling decision is in line with many different works which explicitly separated these states [2, 3], including the study we used as reference when designing our proposed model [4]. It is true that, given the setup described in our paper, D are but a fraction from H. However, having the capability of directly capturing those metrics from different compartments is not only convenient, but also more realistic and useful when visualizing the specific effects of different measures (running our code allows for visual explorations of which node gets infected/hospitalized/dead and when). Having separate states with different transition times could also be exploited in the dynamical/temporal sense to infer noteworthy trends, although such an analysis is beyond the scope of the current paper. Lastly, this setup allows for a node’s behavior within a simulation to change based on its state (for example, one may want to explore a scenario in which hospitalized/recovered people can still ping their contacts through a tracing application, whereas a deceased node would be unable to aid the tracing process).

[1] Moein S, Nickaeen N, Roointan A, et al. Inefficiency of SIR models in forecasting COVID-19 epidemic: a case study of Isfahan. Scientific Reports. 2021;11(1):4725. doi:10.1038/s41598-021-84055-6

[2] Balabdaoui F, Mohr D. Age-stratified discrete compartment model of the COVID-19 epidemic with application to Switzerland. Scientific Reports. 2020;10(1):21306. doi:10.1038/s41598-020-77420-4

[3] Leontitsis A, Senok A, Alsheikh-Ali A, Al Nasser Y, Loney T, Alshamsi A. SEAHIR: A Specialized Compartmental Model for COVID-19. Int J Environ Res Public Health. 2021;18(5). doi:10.3390/ijerph18052667

[4] Di Domenico L, Pullano G, Sabbatini CE, Boëlle P-Y, Colizza V. Impact of lockdown on COVID-19 epidemic in Île-de-France and possible exit strategies. BMC Medicine. 2020;18(1):240. doi:10.1186/s12916-020-01698-4

2. The mechanisms for generating the tracing networks are not described. How are the links chosen, when, and by which mechanism? Is the tracing connected to new infections or not? If not (as this is a mean-field approach), the model might have an intrinsic limitation due to the absence of correlation between the infection process and the network.

We thank the reviewer for pointing out the generation mechanism was not intuitively explained. We have revised Section 2.2 of the paper to address this point. We now describe in more detail how the tracing networks are generated, based on the 2 inter-linked parameters: “degree of overlap” and “uptake”. On one hand, the degree of overlap gets converted to Z_rem via the specified formula, quantity which represents the average number of edges that get randomly removed from the infection network to render a tracing network (this effectively means N * Z_rem / 2 edges get removed, and we have now made this explicit in the revision). On the other hand, the uptake dictates how many nodes should have all their edges removed (again at random). We average over many such tracing scenarios to reduce the potential noise induced by these random processes (hence why we mention in Section 3.1 that we simulate X different networks with Y random seeds). Consequently, the tracing networks are preset, so new infections do not technically influence their structure, but this should not pose a limitation because, in practice, we can assume this ‘static’ structure has been generated from aggregating all the potentially-observed dynamic links for any given node (i.e. a random process corresponding to tracers/apps monitoring the contacts neighborhood of any new detected infections), thus actually correlating the entire procedure to new infections. Given that presumption, the tracing propagation system becomes, in fact, directly influenced by any new detected infection / isolated node, since the latter contributes to the additive probability of encouraging its neighborhood to self-isolate (hence the multi-site nature of our mean-field approach), while the neighborhood itself gets partially observed from the aforementioned pseudo-dynamic process.

That being said, removing the random nature of this operation altogether, and thus actively tying new infections to the tracing view substructure, is indeed an interesting avenue worth exploring (possibly via targeting the testing and the neighborhood exploration processes with reinforcement learning techniques, like [1] - a potential future direction for us as indicated in the revised Section 4), yet this may come at the cost of adding more bias: Currently, we assume contact tracers have a limited time to conduct their research and they do not favor any node when looking for contacts, whereas each individual’s memory constitutes an unpredictable, seemingly-random factor to model. Hence, the simplifying presumption that contacts tend to be missed at random fits these assumptions and should be less biased. This very presumption enables us to view the precomputed static subnetwork as a dynamic aggregate of randomly-removed edges.

[1] Meirom EA, Maron H, Mannor S, Chechik G. How to Stop Epidemics: Controlling Graph Dynamics with Reinforcement Learning and Graph Neural Networks. arXiv:201005313 [cs]. Published online October 26, 2020. Accessed May 15, 2021. http://arxiv.org/abs/2010.05313

3. On the same topic, at line 127 the authors say: "The tracing graphs are usually subsets of the first network." When is this not the case? From the model described in ref 21, I get that tracing might generate "false" contacts, but this is not said at any point in the manuscript.

Thank you for pointing out this inconsistency. It is true that the referenced paper studies such a scenario in more detail. Our model is also capable of simulating “false” contacts, but we have decided not to study such a situation further in this paper since we deemed it less likely to happen in our global pandemic scenario, where contact tracers are generally very well trained and the public is much more aware of who they meet and for how long. We have modified Section 2.2 to better reflect this.

4. The authors validate most of their findings using Erdös-Rényi networks, claiming that "it tends to offer acceptable approximations most of the time" (l146). I must argue that it does not, and in fact the authors themselves then use a scale-free/high clustering model in section 3.4. Why not directly use this one, which is indeed more realistic? They also could use empirical datasets, which are easily found in the community.

We use ER networks as our baseline due to their wide adoption in the epidemic literature (including the studies we used as reference points for the spreading model [1, 2]), as well as their well-behaved epidemic dynamics in the limit, caused by the presence of epidemic thresholds (from [1], the study which introduced the SIRT model we extended, “the epidemic dynamics in SW networks remains qualitatively similar to random graphs, since they possess a well-defined epidemic threshold”). In addition to this, both real networks and ER graphs (under the studied K values) generally exhibit a single giant component, sparse densities, small average shortest paths [3,4]. Even though ER networks make a few unrealistic assumptions about the degree distribution (nodes with too many or too few connections are heavily restricted) and the independence of interactions in closed communities (where some inherent dependence certainly exists), they are still a good tool for modelling infections in randomly mixed populations [5] (e.g. in stores, public transportation etc. [6]). The first few experiments we conducted on the basis of ERs played a pivotal role in exemplifying the overall dynamics induced by different variables in our system, which we preferred to observe (for ourselves) and evoke (to the readers) in the most generalized framework, just as the predecessor papers did in their initial simulations. Indeed, the reviewer was kind to notice that we made a note of the unsuitability such a model may exhibit for social network tasks, hence why we focused the main experiments in the study on a more realistic network model, for which we actually derived our final conclusions and plotted the most representative figures on the efficacy of various contact tracing strategies w.r.t. the peak suppression and the reproduction number. We agree with the reviewer that our original statement about ERs was overly optimistic, and hence we have modified the section accordingly. We have also added Section 3.5 to the paper which details experiments we ran over a real dynamic social network. 

[1] Tsimring LS, Huerta R. Modeling of contact tracing in social networks. Physica A: Statistical Mechanics and its Applications. 2003;325(1):33-39. doi:10.1016/S0378-4371(03)00180-8

[2] Farrahi K, Emonet R, Cebrian M. Epidemic Contact Tracing via Communication Traces. Lambiotte R, ed. PLoS ONE. 2014;9(5):e95133. doi:10.1371/journal.pone.0095133

[3] D. Du, “Social Network Analysis: Lecture 3-Network Characteristics,” Sep. 28, 2016. [Online]. Available: http://www2.unb.ca/~ddu/6634/Lecture_notes/Lec3_network_statistics_handout.pdf

[4] A. L. Barabási, Network Science Random Networks. 2015. [Online]. Available: https://barabasi.com/f/624.pdf

[5] M. J. Keeling and K. T. D. Eames, “Networks and epidemic models,” Journal of The Royal Society Interface, vol. 2, no. 4, pp. 295–307, Sep. 2005, doi: 10.1098/rsif.2005.0051.

[6] M. Abueg et al., “Modeling the combined effect of digital exposure notification and non-pharmaceutical interventions on the COVID-19 epidemic in Washington state,” in medRxiv, Sep. 2020, p. 2020.08.29.20184135. doi: 10.1101/2020.08.29.20184135.

5. Another problem is the temporality of the spreading. Line 155, the authors state: "The time intervals between two state changes of the same kind are assumed to form an exponential distribution". This is very unrealistic. As shown in a wide range of empirical studies, human interactions exhibit bursty behaviour, with distributions of temporal properties having typical heavy-tails. It has been further shown that such properties condition strongly the spread of an epidemic (see for example Lambiotte et al EPJB 86 (2013) 320).

We thank the reviewer for opening up this important discussion on the temporality involved in our simulations, which has prompted us to briefly explain our rationale in Section 2.3 of the revision. Our choice here was motivated by the exponential’s wide adoption in previous epidemiological works relying on SIR [1, 2, 4], as well as evidence of exponentially-distributed numbers of interactions over time observed in 2 reference studies for our work, which analyzed human behavior using tracking devices [2, 3]. We agree with the reviewer that our assumption can be too restrictive, especially surrounding infection times, and more flexible gamma (e.g. Erlang) or Weibull distributions represented via temporal sub-compartments could provide more realistic measurements. However, standard exponentials have been shown to offer acceptable approximations in several cases (e.g. when the mean infection duration is smaller [5], or the mean-generation time is correctly estimated [6]). As to other time intervals involved in the model (e.g. the infectious period), an older study showed that major epidemic persistence and dynamic problems could be introduced in SIR/SEIR simulations when using more realistic gamma distributions, amounting to worse results overall [7]. That is not to say using such distributions should be discouraged in SIR, but a very thoughtful consideration should be put into making this temporality choice nonetheless, carefully weighing the benefits against the potential limitations. We excluded this analysis for the purpose of the current paper, but it is definitely an interesting avenue worth exploring in subsequent studies.

[1] J. Ma, “Estimating epidemic exponential growth rate and basic reproduction number,” Infectious Disease Modelling, vol. 5, pp. 129–141, Jan. 2020, doi: 10.1016/j.idm.2019.12.009.

[2] K. Farrahi, R. Emonet, and M. Cebrian, “Predicting a Community’s Flu Dynamics with Mobile Phone Data,” Vancouver, Canada, Mar. 2015. doi: 10.1145/2675133.2675237.

[3] J. Stehlé et al., “Simulation of an SEIR infectious disease model on the dynamic contact network of conference attendees,” BMC Medicine, vol. 9, no. 1, p. 87, Jul. 2011, doi: 10.1186/1741-7015-9-87.

[4] M. Kröger and R. Schlickeiser, ‘Analytical solution of the SIR-model for the temporal evolution of epidemics. Part A: time-independent reproduction factor’, J. Phys. A: Math. Theor., vol. 53, no. 50, p. 505601, Nov. 2020, doi: 10.1088/1751-8121/abc65d.

[5] E. Vergu, H. Busson, and P. Ezanno, “Impact of the Infection Period Distribution on the Epidemic Spread in a Metapopulation Model,” PLoS One, vol. 5, no. 2, Feb. 2010, doi: 10.1371/journal.pone.0009371.

[6] O. Krylova and D. J. D. Earn, “Effects of the infectious period distribution on predicted transitions in childhood disease dynamics,” J R Soc Interface, vol. 10, no. 84, Jul. 2013, doi: 10.1098/rsif.2013.0098.

[7] A. L. Lloyd, ‘Realistic Distributions of Infectious Periods in Epidemic Models: Changing Patterns of Persistence and Dynamics’, Theoretical Population Biology, vol. 60, no. 1, pp. 59–71, Aug. 2001, doi: 10.1006/tpbi.2001.1525.

6. I suggest the author define the measures they used to quantify the effect of tracing in the text. In particular, what is "peak suppression"? I would recommend a clearer measure, such as a normalised peak reduction (N_(no tracing) - N_tracing)/(N_(no tracing)). Same approach could be used for time of peak.

Thank you for this insightful comment. We have addressed this point by adding a clear mathematical explanation of the peak suppression in the newly added Section 2.4. We would also like to mention that the original purpose of our peak suppression plots was to emphasize the benefit of contact tracing in absolute terms (scrutinized in direct reference to the total population size, which remains fixed at either 1000 or 10000 throughout the study, making it easy for the reader to visualize the full effect). While we agree with the reviewer that normalised charts would offer a valuable alternative view of our simulation data, an unnormalized reporting directly referencing the achieved peak suppression better fits with our initial plan.

7. l203 : The authors select "good" values for tau_r and tau_t, but it seems to me that there is a correlation between the threshold values of tau_r and tau_t. It would be interesting to see heatmaps combining both dependencies.

Thank you very much for this suggestion. We have now added heatmaps of the 2 parameters for different degrees of overlap to the revised Section 3.4.

8. I have a question about the results presented in Fig 5, top right quadrant. It seems that the larger the population, the smaller the maximum infected fraction of the population, with a limit going to 0 as N increases. This would indicate that the model does not generate large cascades of infections, while spreading processes —being analog to percolation— should always reach a non-zero fraction of the population when the infection parameter is above the epidemic threshold. If accurate, doesn't it indicate an intrinsic unrealistic property of the model? It could also be that the parameters chosen for the study in section 3.1 lead to a spreading under the epidemic threshold, but in that case the analysis of the variation induced by population size is irrelevant.

As illustrated by the unnormalized values in the left quadrant, the average of the peak of infection is monotonic on the studied range, and hence, for larger populations, the disease did manage to percolate at an increased rate (in absolute terms) as expected, making the variation analysis hold (at least on this very range). In spite of this, as indicated in the right quadrant and correctly pointed out by the reviewer, the peak clearly decreases in relative importance w.r.t. the total population (and the above-named monotony is not enough to ensure a non-zero relative average in the limit). Even so, the purpose of this experiment was not to study the dynamics in the limit, but rather to check if the variability across runs scales with the number of nodes (as initially suspected), and to settle on a sensible range of values for the population size (which would not be plagued by large variances, at least relative to that size). Indeed, as the reviewer correctly intuited, the limiting behavior of the percolation process is the result of analyzing a scenario in which both tau_t and tau_r are marginally larger than the infection rate beta (0.1 against 0.0791). Should the uptake have been 100% and considering that both the infection and contact tracing “spread” in a similar fashion, the simulation would have been deterministically contained (and below the epidemic threshold with a greater probability). However, given that the uptake in this experiment is set to 50%, this is now not always the case, as some instances do get quickly contained, while others continue to percolate for longer periods, depending on the network subsetting seed. It is also worth noting here that Section 3.1 reports on the variance of peaks, rather than overall epidemic sizes. While tracing can indeed be effective at “flattening the curve”, this does not guarantee the final size actually follows the same trend (or has the same decaying shape in the limit). 

On a side note, our choice of studying \\tau_t=\\tau_r=0.1 here stemmed from empirical observations of this particular assignment resulting in an “average” efficacy w.r.t. preventing full-blown outbreaks, across many different parameter configurations (possibly due to the aforementioned marginal difference compared to the transmission rate). As reported in the paper, this assignment belongs to the higher-mid range of our initial setups, but also lies within the lower-mid range in the main experiments.

9. The study would be much stronger if the model was explored and validated for a larger range of spreading parameters. Only two values for p_a and one for p_h are considered. I understand that the authors have used estimated values from the empirical studies, but since the model for interactions is not realistic, it might be that these values are not suitable to reproduce the desired spreading properties (see my previous comment). Similarly, only two values for K are considered. I understand that the model is complex and the parameter space is huge, but the current results merely rely on eye-balling "relevant" values from a minimal set of tries. Deriving quantitative, meaningful results from such an evaluation of "proper" values for the parameters seems too optimistic.

As the reviewer kindly noted, the parameter space is indeed very large, and it is very difficult to find the most realistic values here. We concentrated our experiments on the most agreed-upon values, which were previously derived by trusted institutions and used to inform real governmental policies (by the CDC [1], by the The French National Institute of Health and Medical Research [2]). Our compartmental model was deliberately designed to be “in line” with [2], and therefore we were very composed with parameter changes (in fact, modifications were conducted only based on estimates from other trusted sources). This makes our results meaningful for the scenario in question. That being said, we consider our main contribution to be the modelling technique itself (aspect reflected in our decision to submit to this particular journal), and hence the range of the parameters was constrained to the most sensible values for presentation. A wider range of tau_r and tau_t values was, however, studied in our experiments, some significantly below and above the epidemic threshold, with the final ranges reported in the final piece being indicative of all the observed trends. We agree with the reviewer that a more thorough analysis of the parameters should be carried away in future work, especially in relation with K, and this has been marked accordingly in Section 4 (Conclusions and future work).

[1] M. A. Johansson et al., “SARS-CoV-2 Transmission From People Without COVID-19 Symptoms,” JAMA Netw Open, vol. 4, no. 1, p. e2035057, Jan. 2021, doi: 10.1001/jamanetworkopen.2020.35057.

[2] L. Di Domenico, G. Pullano, C. E. Sabbatini, P.-Y. Boëlle, and V. Colizza, “Impact of lockdown on COVID-19 epidemic in Île-de-France and possible exit strategies,” BMC Medicine, vol. 18, no. 1, p. 240, Jul. 2020, doi: 10.1186/s12916-020-01698-4.

10. Manual tracing and digital tracing seem to be a sort of redundant system, to increase the global tracing efficiency. Shouldn't the use of both be equivalent to having a single, higher value of either digital tracing or manual tracing?

The two tracing systems operate on different assumptions (please refer back to Section 3.4). Namely, both conduct tracing over different networks (a sensible assumption since not all interviewees will have a COVID tracing application to notify neighbors, and there may be discrepancies between what each service “sees”). In manual tracing, the overlap controls missed contacts, while in digital tracing both an overlap (inadequate use) and an uptake rate can influence the topology (the latter does not make sense in the manual tracing case). Finally, there is an inherent delay of tracing a contact in the interview-based approach (controlled by differences in \\tau_t rates; assumed to be equal to two days in our study, but with direct support in the code for varying that delay).

11. I would appreciate that the authors discuss two features from Fig 7:

Thank you for the questions below. First, we would like to note that we have changed Fig 7 in the revision to match the given caption, since in the initial submission we have imported by mistake an older version of that figure (with less scenarios depicted). In the new figure, the 4 quadrants visible in the originally-submitted picture are all placed on the right column. We have also removed the case with overlap=0 since this theoretically corresponds to the same scenario as \\tau_t=0. That being said, the questions below still hold since this figure update is based on the same data and scenarios.

11.1. How can contact tracing lead to situations in which the outbreak lasts longer?

First of all, we must distinguish all cases from the first row, which we regard as giving mostly noisy results of the contact tracing process due to the ineffective value of \\tau_t. In some of the more-meaningful configurations, very small values of the overlap (<.33) still manage to fall within the “noise region” because \\tau_t is not large enough to make their effect distinguishable from the noise induced by the inherent stochasticity of the simulations. Such a poor representation of the infection network can also lead to situations in which only incorrect regions of the graph get isolated (i.e. many false positives), which in turn can cause the epidemics to gain momentum at later stages (e.g. an obvious example is overlap=0.11 in the third quadrant). A brief explanation has been added to the revision of Section 3.2.

11.2. What generates the bimodality of some curves?

The effect of multiple peaks has also been noted for the SIR formulation in [1]: “with larger tracing effort, the epidemic is reduced significantly rapidly, leaving a great deal of the population susceptible for a second peak of infections[…] This is due to the contact tracing becoming so effective that the number of cases drops rapidly, resulting in tracing becoming less effective. Note, this effect is much more attenuated in the dual network case” (the “dual network case” refers to a scenario with a single contact tracing network in which the overlap is very small: 0.08). Indeed, in our case this bimodality is a rarer phenomenon which only affects a few of the simulations that exhibit large values of the overlap.

[1] Farrahi K, Emonet R, Cebrian M. Epidemic Contact Tracing via Communication Traces. Lambiotte R, ed. PLoS ONE. 2014;9(5):e95133. doi:10.1371/journal.pone.0095133

12. Furthermore, I have some minor issues with the manuscript:

- I would advise for replacing ref 21 with "R. Huerta, L. Tsimring, Phys. Rev. E 66 (2002) 056 115". I am not an expert on multi-site mean-field models, and I found that the latter reference contains a much clearer description of the approach than the one given by the authors.

- l196 : please move the reference to Fig 7 at the relevant location ; the current one is a sentence about Fig 6

We thank the reviewer for spotting these minor issues. Indeed, the initial paper by Huerta and Tsimring includes more details on multi-site mean-field models, but the latter features more experiments and contains further conclusions derived from them. This is why we decided to keep both references in the revision. As to Fig 7, as mentioned above, the caption was in fact correct but we imported an older version of that figure onto the system. This has now been amended.

Replies to reviewer #2:

1. The conclusions of the simulation study are weakened by the fact that they rely solely on simulated contact networks which lack many realistic features of real world contact patterns. I think the model in itself and the present study merit publication on their own, but it would be a big plus if the authors could motivate that the networks’ parameters are realistic and that the missing features do not change the conclusions, and/or perform simulations on empirical contact networks.

We thank the reviewer for this important suggestion, and we agree that the study would benefit from showcasing the impact of contact tracing on real social networks. To that end, we have added Section 3.5 (with background explanations added to Section 2.3) where we explore a real data-driven scenario.

2. The model has a lot of different variables and parameters and it is difficult to keep track of them all. It would be very helpful to list all of them in a table, similar to Table 1. 

Namely: \\Gamma, K, r, t_rem. It might also be helpful to list the different compartments (states) of the model.

Thank you for this suggestion. A table detailing the network parameters used in our simulations has been added to the revision. A diagram of the compartments together with the available transitions has been supplied in Fig 1, with the caption detailing their meaning.

3. In the same manner, it would make the manuscript easier to read if the names, or short definitions, were given for each of the parameters when their ranges considered are listed in each subsection of the Results, e.g., on lines 187 and 188 recall that \\Gamma is the degree of overlap, \\tau_t the contact tracing rate, etc.

Thank you for this thoughtful suggestion. The names of the different parameters have now been made apparent at the beginning of each section.

4. Add values of all relevant parameters to figure captions. This will make the figures easier to read.

We thank the reviewer for this suggestion. The values of the relevant parameters have now been added to the figure captions in the revised study.

5. It is unintuitive to use S_i for general states and S for the susceptible state. I suggest the authors use another symbol to denote a general state, e.g. X_i.

We have amended the table according to this suggestion, using X to refer to different states.

6. The fact that the relative error is ~3% for networks of size N=1000 and ~1% for N=10000 is corresponds to a relative error of 1/\\sqrt{N}, consistent with central limit type arguments for a well mixed population. The result may not hold however in a structured population or near a critical point and thus cannot necessarily be extrapolated to general conditions.

We thank the reviewer for this insightful comment. We have now made a note of the limitations our variance analysis has.

7. The authors refer to some parameters of their model as “hyperparameters”. This term is generally reserved for parameters of an inference procedure, not a model, e.g., of the learning procedure in machine learning or of prior distributions in Bayesian inference.

The term hyperparameters was chosen to distinguish network parameters from compartmental model parameters. We have now modified the nomenclature accordingly to avoid any confusion.

8. The authors compare the uptake rate in the simulations to the digital tracing adoption rates reported for different countries. However, the adoption rates cited are typically calculated as numbers of download relative to the population, which is not the same as the fraction of population using the app correctly.

The adoption rates cited are estimated from application downloads, which can actually represent significant overestimations of the true uptake (this is now apparent in Section 3.4 of the revised text). As to modelling the fraction of the population using the app correctly per se, our simulations should technically rely on the degree of overlap \\Gamma instead of the uptake r. The former, however, was kept to 1 in the digital tracing network, as was correctly observed by the reviewer in the next question.

8. In Section 3.4, if I understood correctly, the overlap between edges in the digital contact tracing network and the true contact network is assumed to be perfect, while the uptake in the manual tracing network is assumed to be 100%. Both of these assumptions seem overly optimistic to me, e.g., the manual contact tracing system will saturate at a given prevalence as was seen in most western countries, while digital contacts are only proxies for real (possibly disease transmitting) contacts.

Indeed, we are only varying the overlap for manual tracing and the uptake for digital tracing, but the two parameters are interlinked within a single network (varying one can directly impact the other). That being said, the “uptake” in manual tracing does not have a clear real-life correspondent, unless we are referring to memory-impaired individuals who are unable to identify any of their contacts (but for the purpose of our study we assumed such special circumstances can be covered by the randomness of link removal attributed to diversifying the degree of overlap). In contrast, varying the “overlap” in digital tracing corresponds to inadequate usage patterns of the application, and was made entirely possible in our codebase. However, showcasing the effects of sampling the entire parameter grid becomes combinatorically harder, so we decided to illustrate only a targeted subset of the scenarios our model is actually capable of running. 

8. Please corroborate the claim (on lines 144-146) that the Erdos-Renyi graph model tends to offer acceptable approximations most of the time.

From [1], the study which introduced the SIRT model we extended, “the epidemic dynamics in SW networks remains qualitatively similar to random graphs, since they possess a well-defined epidemic threshold”. In addition to this, both real networks and ER graphs (under the studied K values) generally exhibit a single giant component, sparse densities, small average shortest paths [3,4]. Even though ER networks make a few unrealistic assumptions about the degree distribution (nodes with too many or too few connections are heavily restricted) and the independence of interactions in closed communities (where some inherent dependence must actually exist), they are still a good tool for modelling infections in randomly mixed populations [5] (e.g. in stores, public transportation etc. [6]). That being said, we do agree our original statement is overly optimistic, and therefore we amended our formulation in the revision.

[1] Tsimring LS, Huerta R. Modeling of contact tracing in social networks. Physica A: Statistical Mechanics and its Applications. 2003;325(1):33-39. doi:10.1016/S0378-4371(03)00180-8

[2] Farrahi K, Emonet R, Cebrian M. Epidemic Contact Tracing via Communication Traces. Lambiotte R, ed. PLoS ONE. 2014;9(5):e95133. doi:10.1371/journal.pone.0095133

[3] D. Du, “Social Network Analysis: Lecture 3-Network Characteristics,” Sep. 28, 2016. [Online]. Available: http://www2.unb.ca/~ddu/6634/Lecture_notes/Lec3_network_statistics_handout.pdf

[4] A.-L. Barabási, Network Science Random Networks. 2015. [Online]. Available: https://barabasi.com/f/624.pdf

[5] M. J. Keeling and K. T. D. Eames, “Networks and epidemic models,” Journal of The Royal Society Interface, vol. 2, no. 4, pp. 295–307, Sep. 2005, doi: 10.1098/rsif.2005.0051.

[6] M. Abueg et al., “Modeling the combined effect of digital exposure notification and non-pharmaceutical interventions on the COVID-19 epidemic in Washington state,” in medRxiv, Sep. 2020, p. 2020.08.29.20184135. doi: 10.1101/2020.08.29.20184135.

8. It seems like an error that symbols in Table 1 are in boldface.

This was done on purpose to highlight the afferent symbols, but in the new version of the table we removed this styling.

9. The authors may want to discuss the implications of the crossover effects seen in Fig. 10, i.e., the final outbreak size can be larger for higher uptake.

A discussion on the rare “crossover effects” has been added to the revised manuscript in Section 3.3. 

10. Can the authors comment on how realistic the required values of \\Gamma and \\tau_t for combined tracing to be effective alone are? Have the values of these parameters been estimated empirically?

The parameter values we indicated as being “effective” have been estimated empirically through inspecting the simulation results. Unfortunately, it is a rather challenging task to find real-world estimates of either of these two parameters. This is why we conduct an extensive study of the trends emerged from running simulations over a large range of values. The UK government SAGE has recommended that at least 80% of close contacts be reached for the system to be deemed effective – overlap=0.8: https://www.health.org.uk/news-and-comment/charts-and-infographics/nhs-test-and-trace-performance-tracker. This percentage is highly influenced, however, by the recalled contacts (only 74% provided at least one close contact in the last month) and the possibility to reach those contacts (84% identified and asked to isolate in the last month). This would bring the actual overlap to a value significantly lower than 0.8 (the quoted value of 0.5 being reasonable to assume as possible to achieve).

11. In Fig. 5 add description of what symbols and boxes and whiskers represent.

Line 74: Change “SIR” to “The SIR process”

Line 80: add abbreviation “(Inserm)”

Line 121: “onto” → “on”

Lines 140-141: should “scales with” be replaced by “is proportional to”?

Line 205: Specify what is meant by “meaningful results”.

Line 212: “N” → “small N”.

Line 302: “encapsulated” → “model’s”

Thank you for these helpful suggestions. We have accommodated these in the revision.

---

## [Decision Letter · Decision Letter 1]

20 Sep 2021

PONE-D-21-07031R1Modelling digital and manual contact tracing for COVID-19. Are low uptakes and missed contacts deal-breakers?PLOS ONE

Dear Dr. Rusu,

Thank you for submitting your manuscript to PLOS ONE.  We invite you to submit a revised version one more time.

Immediate acceptance is possible afterwards. Please focus on answering Second Reviewer's question about the waiting time distribution and implement their minor comments.

We look forward to receiving your revised manuscript.

Kind regards,

Ivan Kryven

Academic Editor

PLOS ONE

Journal Requirements:

Reviewers' comments:

Reviewer's Responses to Questions

**Comments to the Author**

1. If the authors have adequately addressed your comments raised in a previous round of review and you feel that this manuscript is now acceptable for publication, you may indicate that here to bypass the “Comments to the Author” section, enter your conflict of interest statement in the “Confidential to Editor” section, and submit your "Accept" recommendation.

Reviewer #2: All comments have been addressed

Reviewer #3: All comments have been addressed

2. Is the manuscript technically sound, and do the data support the conclusions?

Reviewer #2: Yes

Reviewer #3: Yes

3. Has the statistical analysis been performed appropriately and rigorously? 

Reviewer #2: N/A

Reviewer #3: Yes

4. Have the authors made all data underlying the findings in their manuscript fully available?

Reviewer #2: Yes

Reviewer #3: Yes

5. Is the manuscript presented in an intelligible fashion and written in standard English?

Reviewer #2: Yes

Reviewer #3: Yes

6. Review Comments to the Author

Reviewer #2: I apologize for the delay in submitting my report.

The authors have addressed all my previous criticism in their revision. I am happy to recommend the publication of their paper in PLOS ONE.

The authors have added a discussion of the modeling assumption that state transitions take place with constant rates (exponentially distributed waiting times). Which is great since this common assumption is often not satisfied in empirical data. However, they discuss only the case where waiting time distributions are less skewed than exponentials. Though they may also be more skewed, notably due to intercontact times and edge weights following heavy-tailed distributions as is often the case in physical proximity networks often (see e.g. Starnini et al. “Modeling human dynamics of face-to-face interaction networks” PRL 2013).

One has to read Eqs. 1 and 2 to understand how $\\Gamma$ and $r$ are defined, while the definitions of $Z_{rem}$ and $N_{utn}$ are clear from the text. I suggest moving the first part of Eqs. 1 and 2 (before the $\\Rightarrow$ sign) up to where $\\Gamma$ and $r$ are introduced. The second parts of Eqs. 1 and 2 may be removed as they are obtained by a simple arithmetic inversion of the first parts, or the authors may keep them at their current place.

Change: ‘contacts network’ to ‘contact network’ on page 6 (no line numbering) and on page 7, line 137.

Change: ‘newly isolated’ to ‘isolated’ on page 6.

Change: ‘testing regimes’ to ‘testing regime’ in Fig. 7 caption.

Change: ‘$tau_r$’ to ‘$\\tau_r$’ in line 252.

Change ‘First aspect’ to ‘The first aspect’ in line 304.

Change: ‘experiments’ to ‘simulations’ in line 371.

Reviewer #3: The manuscript presents a compartmental model that can explicitly capture the manual/digital contact tracing and study contact tracing strategies in various situations. Overall, I believe that the work mostly satisfies the publication criteria of PLOS ONE, providing a solid study of the proposed model.

However, I still have two comments. First, as other reviewers mentioned, the network structure and temporal dynamics are somewhat overlooked in the paper. Although I would not argue that the authors should perform extra simulations, I think it is important to acknowledge it more thoroughly in the discussion. It has been recognized that super-spreading is a rather universal characteristic of many epidemics [1] and COVID-19 is argued to be driven primarily by such super-spreading events. Furthermore, recent studies (e.g., [2]) have shown that when spreading is driven by such super-spreading events, the details of contact tracing implementation may matter a lot. In this context, I believe that the paper should expand the discussion to provide a better context to the readers.

Second, I think the plots can be improved a lot by carefully choosing colors and by limiting the number of lines/objects that each figure shows. Many figures have numerous (~10) lines with random colors associated with each line, making them very difficult to parse. I believe that most of the figures will not lose much information by reducing the number of lines to ~5. In addition, as each of these lines show a range of parameter values (i.e., they can be ordered), a linear colormap (e.g., sampling colors across the "viridis" colormap) would make them much easier to read. Also, the heatmap figure uses green-to-red colormap, which can be understood by a significant fraction of population who has colorblindness. Furthermore, the colormap used introduces an arbitrary cut-off point (between 750 and 1000) that introduces an artifact. Again, I believe that a linear, perceptually uniform colormap should be used here. Although this is probably not "critical" regarding PLOS ONE's publication criteria, I believe that this simple improvement in the figures will make the paper much more accessible.

[1] Lloyd-Smith, J. O., Schreiber, S. J., Kopp, P. E. & Getz, W. M. Superspreading and the effect of individual variation on disease emergence. Nature 438, 355–359 (2005).

[2] Kojaku, S., Hébert-Dufresne, L., Mones, E., Lehmann, S., & Ahn, Y. Y. (2021). The effectiveness of backward contact tracing in networks. Nature Physics, 17(5), 652-658.

7. PLOS authors have the option to publish the peer review history of their article (what does this mean?). If published, this will include your full peer review and any attached files.

Reviewer #2: No

Reviewer #3: No

---

## [Author Response · Author response to Decision Letter 1]

26 Oct 2021

Replies to editor

We reverified our References list to ensure no retracted papers got cited. At the same time, we added a few more references to support our discussion on waiting times and network structures, in light of the suggestions made by the reviewers. In addition, we supplemented the information provided for references 5, 6, 10, 16 (previously 15), 17 (previously 16) with links to the corresponding cited web resources. Finally, we modified references 18 (previously 17) and 45 (previously 65) to reflect these papers’ recent publication.

All our figures have been verified using the PACE system, and no issues were reported, except for Fig8.eps in which the tool detected bitmap images with resolution < 300 DPI. We have inspected the apparently-problematic figure but found no issues with its rendering. Please also note that most of our figures were regenerated using a different colormap in light of the comments made by Reviewer #3, and adequate margins have now been added.

Replies to Reviewer #2

1. The authors have added a discussion of the modeling assumption that state transitions take place with constant rates (exponentially distributed waiting times). Which is great since this common assumption is often not satisfied in empirical data. However, they discuss only the case where waiting time distributions are less skewed than exponentials. Though they may also be more skewed, notably due to intercontact times and edge weights following heavy-tailed distributions as is often the case in physical proximity networks often (see e.g. Starnini et al. “Modeling human dynamics of face-to-face interaction networks” PRL 2013).

We thank the reviewer for pointing out that intercontact times may often be gracefully modelled via heavier-tailed distributions, such as power laws. We have now added a comment on this aspect in the manuscript. The decision we took for our contact tracing model was based on the findings of 2 cohort studies that reported exponential decays in the number of interaction events over time, while also having in mind the fact that this is a very common assumption in the epidemiological literature. The work of Starnini et al. [1] found power laws to be excellent approximators for the contact duration, but a power law fit to waiting times was found to be moderately less optimal, as the authors remark themselves. We would also like to emphasize that the figures presented in the aforementioned study utilize a log-log scale, which may sometimes hide exponential trends. In fact, the authors of this paper do not provide the parameters for the fitted power law to the intercontact times (although they do so for contact duration), so it is harder to judge the extent of the inferred exponent. At the same time, the fitted curve exhibits a concave curvature in the log-log plot, which may actually correspond to a roughly-exponential trend if translated to a linear scale. Sadly, no comparison with an exponential fit is provided in this text either. We acknowledge, however, that some interaction datasets do feature more skewed waiting times distributions than Gamma, and hence we believe that exploring the impact of utilizing longer-tailed distributions is an interesting avenue worth exploring in future endeavors.

[1] Starnini et al. “Modeling human dynamics of face-to-face interaction networks” PRL 2013

2. One has to read Eqs. 1 and 2 to understand how $\\Gamma$ and $r$ are defined, while the definitions of $Z_{rem}$ and $N_{utn}$ are clear from the text. I suggest moving the first part of Eqs. 1 and 2 (before the $\\Rightarrow$ sign) up to where $\\Gamma$ and $r$ are introduced. The second parts of Eqs. 1 and 2 may be removed as they are obtained by a simple arithmetic inversion of the first parts, or the authors may keep them at their current place.

Thank you for suggesting this easier-to-comprehend rephrasing and reordering of Section 2.2 and its equations. The manuscript has been amended to include in-line equations for all the network parameters involved, while full equations for $N_{utn}$ and $N_{ute}$ have been provided below.

3. Change: ‘contacts network’ to ‘contact network’ on page 6 (no line numbering) and on page 7, line 137. Change: ‘newly isolated’ to ‘isolated’ on page 6.

Change: ‘testing regimes’ to ‘testing regime’ in Fig. 7 caption.

Change: ‘$tau_r$’ to ‘$\\tau_r$’ in line 252.

Change ‘First aspect’ to ‘The first aspect’ in line 304.

Change: ‘experiments’ to ‘simulations’ in line 371.

Thank you for signaling these difficult-to-spot phrasing issues and mistakes. These suggestions have now been accommodated in the final manuscript.

Replies to Reviewer #3

1. First, as other reviewers mentioned, the network structure and temporal dynamics are somewhat overlooked in the paper. Although I would not argue that the authors should perform extra simulations, I think it is important to acknowledge it more thoroughly in the discussion. It has been recognized that super-spreading is a rather universal characteristic of many epidemics [1] and COVID-19 is argued to be driven primarily by such super-spreading events. Furthermore, recent studies (e.g., [2]) have shown that when spreading is driven by such super-spreading events, the details of contact tracing implementation may matter a lot. In this context, I believe that the paper should expand the discussion to provide a better context to the readers.

[1] Lloyd-Smith, J. O., Schreiber, S. J., Kopp, P. E. & Getz, W. M. Superspreading and the effect of individual variation on disease emergence. Nature 438, 355–359 (2005).

[2] Kojaku, S., Hébert-Dufresne, L., Mones, E., Lehmann, S., & Ahn, Y. Y. (2021). The effectiveness of backward contact tracing in networks. Nature Physics, 17(5), 652-658.

We thank the reviewer for pointing out that our discussion on superspreading, a very important mechanism underpinning the spread of COVID-19, was minimal in the previous submission. As such, we have now included a larger paragraph on superspreading, preferential attachment and small-world networks (see Section 2.3). In the same section, we have also commented on why we believe our contact tracing modelling technique remains suitable in the face of challenges like superspreading in scale-free networks or large clustering coefficients in small-world graphs, taking into consideration the findings of Kojaku et al. [1] and Tsimring and Huerta [2], respectively, as well as the similarities between our modelling technique and theirs. 

[1] Kojaku, S., Hébert-Dufresne, L., Mones, E., Lehmann, S., & Ahn, Y. Y. (2021). The effectiveness of backward contact tracing in networks. Nature Physics, 17(5), 652-658.

[2] Tsimring LS, Huerta R. Modeling of contact tracing in social networks. Physica A: Statistical Mechanics and its Applications. 2003;325(1):33-39. doi:10.1016/S0378-4371(03)00180-8

2. Second, I think the plots can be improved a lot by carefully choosing colors and by limiting the number of lines/objects that each figure shows. Many figures have numerous (~10) lines with random colors associated with each line, making them very difficult to parse. I believe that most of the figures will not lose much information by reducing the number of lines to ~5. In addition, as each of these lines show a range of parameter values (i.e., they can be ordered), a linear colormap (e.g., sampling colors across the "viridis" colormap) would make them much easier to read. Also, the heatmap figure uses green-to-red colormap, which can be understood by a significant fraction of population who has colorblindness. Furthermore, the colormap used introduces an arbitrary cut-off point (between 750 and 1000) that introduces an artifact. Again, I believe that a linear, perceptually uniform colormap should be used here. Although this is probably not "critical" regarding PLOS ONE's publication criteria, I believe that this simple improvement in the figures will make the paper much more accessible.

Indeed, the original color schemes used for our plots were sometimes counterintuitive and probably difficult to comprehend for people with color blindness. To fix this issue, we have regenerated all our charts, including the heatmaps, using the linear colormap ‘inferno’. What is more, we have also reduced the number of lines in all the overlap/uptake figures to a maximum of 7 different rates, thus making them more comprehendible.

---

## [Editor Report · Decision Letter 2]

2 Nov 2021

Modelling digital and manual contact tracing for COVID-19. Are low uptakes and missed contacts deal-breakers?

PONE-D-21-07031R2

Dear Dr. Rusu,

We’re pleased to inform you that your manuscript has been judged scientifically suitable for publication and will be formally accepted for publication once it meets all outstanding technical requirements.

Kind regards,

Ivan Kryven

Academic Editor

PLOS ONE
---

## [Editor Report · Acceptance letter]

8 Nov 2021

PONE-D-21-07031R2 

Modelling digital and manual contact tracing for COVID-19 Are low uptakes and missed contacts deal-breakers? 

Dear Dr. Rusu:

I'm pleased to inform you that your manuscript has been deemed suitable for publication in PLOS ONE. Congratulations! Your manuscript is now with our production department. 

Kind regards, 

on behalf of

Dr. Ivan Kryven 

Academic Editor

PLOS ONE